# Temporal integration is a robust feature of perceptual decisions

**Alexandre Hyafil[1,2]\*, Jaime de la Rocha[3], Cristina Pericas[3], Leor N Katz[4], Alexander C Huk[5], Jonathan W Pillow[2]**

[1]Centre de Recerca Matemàtica, Bellaterra, Spain; [2]Princeton Neuroscience Institute, Princeton University, Princeton, United States; [3]Institut d'Investigacions Biomèdiques August Pi i Sunyer (IDIBAPS), Barcelona, United States; [4]Laboratory of Sensorimotor Research, National Eye Institute, National Institutes of Health, Bethesda, United States; [5]Fuster Laboratory for Cognitive Neuroscience, Departments of Psychiatry & Biobehavioral Sciences and Ophthalmology, UCLA, Los Angeles, United States

**\*For correspondence:**
alexandre.hyafil@gmail.com

**Competing interest:** The authors declare that no competing interests exist.

**Abstract** Making informed decisions in noisy environments requires integrating sensory information over time. However, recent work has suggested that it may be difficult to determine whether an animal's decision-making strategy relies on evidence integration or not. In particular, strategies based on extrema-detection or random snapshots of the evidence stream may be difficult or even impossible to distinguish from classic evidence integration. Moreover, such non-integration strategies might be surprisingly common in experiments that aimed to study decisions based on integration. To determine whether temporal integration is central to perceptual decision-making, we developed a new model-based approach for comparing temporal integration against alternative 'non-integration' strategies for tasks in which the sensory signal is composed of discrete stimulus samples. We applied these methods to behavioral data from monkeys, rats, and humans performing a variety of sensory decision-making tasks. In all species and tasks, we found converging evidence in favor of temporal integration. First, in all observers across studies, the integration model better accounted for standard behavioral statistics such as psychometric curves and psychophysical kernels. Second, we found that sensory samples with large evidence do not contribute disproportionately to subject choices, as predicted by an extrema-detection strategy. Finally, we provide a direct confirmation of temporal integration by showing that the sum of both early and late evidence contributed to observer decisions. Overall, our results provide experimental evidence suggesting that temporal integration is an ubiquitous feature in mammalian perceptual decision-making. Our study also highlights the benefits of using experimental paradigms where the temporal stream of sensory evidence is controlled explicitly by the experimenter, and known precisely by the analyst, to characterize the temporal properties of the decision process.

## Editor's evaluation

This manuscript tests an important assumption about how sensory information is processed and used to guide motor choices. The widely held assumption is that sensory-motor circuits are capable of integrating evidence, but the validity and generality of this 'principle' have been recently questioned by studies suggesting that other computational operations may lead to similar psychophysical results, mimicking integration without actually performing it. This study makes a compelling case that the integration assumption was likely correct all along and that the model mimicry can be easily disambiguated by using appropriate sensory stimuli and task designs that permit rigorous analyses.

## Introduction

Perceptual decision-making is thought to rely on the temporal integration of noisy sensory information on a timescale of hundreds of milliseconds to seconds. Temporal integration corresponds to summing over time the evidence provided by each new sensory stimulus, and optimizes perceptual judgments in face of noise (*Bogacz et al., 2006*; *Gold and Shadlen, 2007*). A perceptual decision can then be made on the basis of this accumulated evidence, either as some threshold on accumulated evidence is reached, or if some internal or external cue signals the need to initiate a response.

Although many behavioral and neural results are consistent with this integration framework, temporal integration is a feature that has often been taken for granted rather than explicitly tested. Recently, the claim that standard perceptual decision-making tasks rely on (or even frequently elicit) temporal integration has been challenged by theoretical results showing that non-integration strategies can produce behavior that carries superficial signatures of temporal integration (*Stine et al., 2020*). These signatures include the relationship between stimulus difficulty, stimulus duration, and behavioral accuracy, the precise temporal weighting of sensory information on the decisions, and the patterns of reaction times.

Here, we propose new analytical tools for directly assessing integration and non-integration strategies from fixed- or variable-duration paradigms where, critically, the experimenter controls the fluctuations in perceptual evidence over time within each trial (discrete-sample stimulus, or DSS). By leveraging these controlled fluctuations, our methods allow us to make direct comparisons between integration and non-integration strategies. We apply these tools to assess temporal integration in data from monkeys, humans, and rats that performed a variety of perceptual decision-making tasks with DSS. Applying these analyses to these behavioral datasets yields strong evidence that perceptual

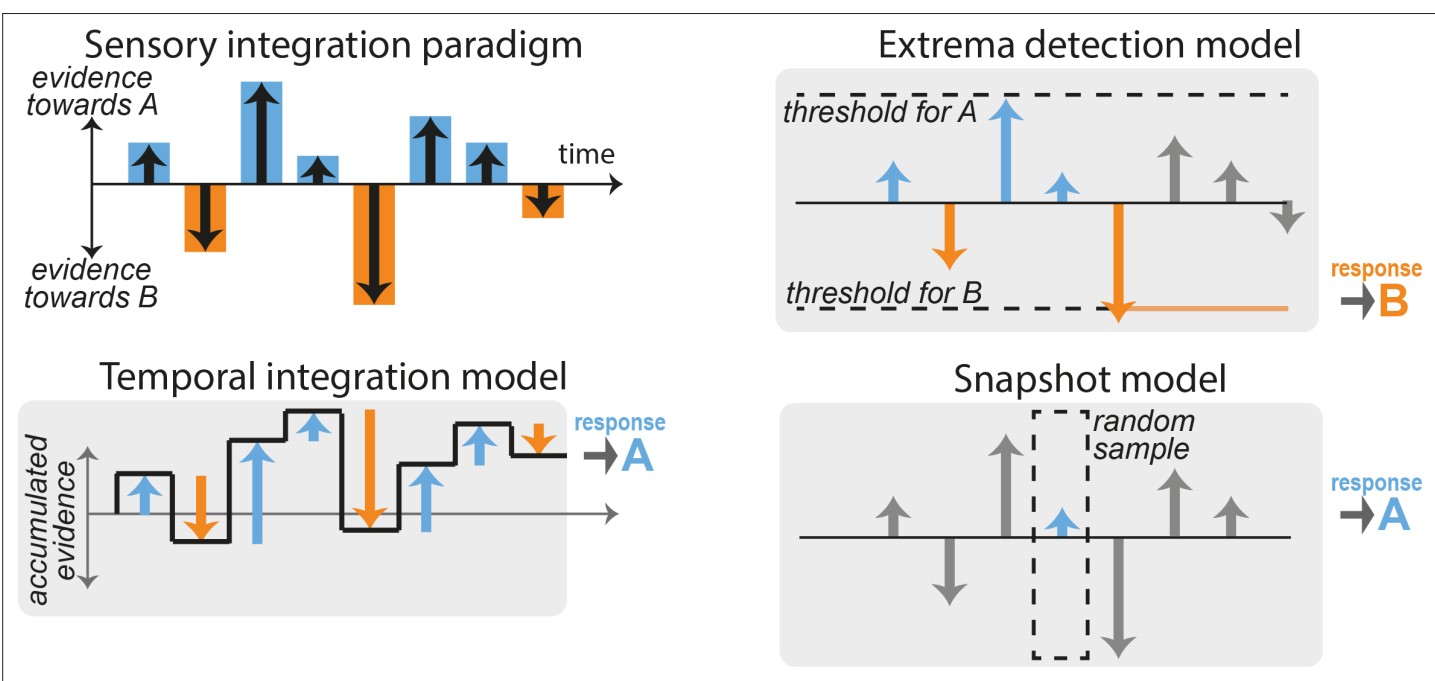

**Figure 1.** Integration and non-integration models for performing sensory discrimination tasks.
(**A**) Schematic of a typical fixed-duration perceptual task with discrete-sample stimuli (DSS). A stimulus is composed of a discrete sequence of *n* samples (here, *n* = 8). The subjects must report at the end of the sequence whether one specific quality of the stimulus was 'overall' leaning more toward one of two possible categories A or B. Evidence in favor of category A or B varies across samples (blue and orange bars). (**B**) Temporal integration model. The relative evidence in favor of each category is accumulated sequentially as each new sample is presented (black line), resulting in temporal integration of the sequence evidence. The choice is determined by the end point of the accumulation process: here, the overall evidence in favor of category A is positive, so response A is selected. (**C**) Extrema-detection model. A decision is made whenever the instantaneous evidence for a given sample (blue and orange arrows) reaches a certain fixed threshold (dotted lines). The selected choice corresponds to the sign of the evidence of the sample that reaches the threshold (here, response B). Subsequent samples are ignored (gray bars). (**D**) Snapshot model. Here, only one sample is attended. Which sample is attended is determined in each trial by a stochastic policy. The response of the model simply depends on the evidence of the attended sample. Other samples are ignored (gray bars). Variants of the model include attending *K* > 1 sequential samples.

decision-making tasks in all three species rely on temporal integration. Temporal integration, a critical element of many major theories of perception at both the neural and behavioral levels, is indeed a robust and pervasive aspect of mammalian behavior. Our results also illuminate the power of targeted stimulus design and statistical analysis to test specific features of behavior.

## Results

### Integration and non-integration models

In a typical perceptual evidence-integration experiment (*Figure 1A*), an observer is presented in each trial with a time-varying stimulus and must report which of two possible stimulus categories it belongs to. Typical examples include judging whether a dynamic visual stimulus is moving leftwards or rightwards (*Yates et al., 2017*; *Katz et al., 2015*); whether the orientation of a set of gratings is more aligned with cardinal or diagonal directions (*Wyart et al., 2012*) whether a combination of tones is dominated by high or low frequencies (*Morillon et al., 2014*; *Hermoso-Mendizabal et al., 2020*; *Znamenskiy and Zador, 2013*); which of two acoustic streams is more intense or dense (*Brunton et al., 2013*; *Pardo-Vazquez et al., 2019*; *Cisek et al., 2009*). Such paradigms have been used extensively in humans, nonhuman primates, and rodents. Here, we focus on experiments in which observers report their choice at the end of a period whose duration is controlled by the experimenter (*Kiani and Shadlen, 2009*; *Wyart et al., 2012*; *Brunton et al., 2013*; *Raposo et al., 2012*), in contrast to so-called 'reaction time' tasks, in which the observer can respond after viewing as brief a portion of the stimulus as they wish (*Roitman and Shadlen, 2002*; *Znamenskiy and Zador, 2013*; *Pardo-Vazquez et al., 2019*; *Hermoso-Mendizabal et al., 2020*).

Moreover, we focus on experimental paradigms in which the sensory evidence in favor of each category arrives in a sequence of discrete *samples*. Samples can correspond to motion pulses (*Yates et al., 2017*), individual gratings (*Wyart et al., 2012*), acoustic tones (*Morillon et al., 2014*; *Hermoso-Mendizabal et al., 2020*; *Znamenskiy and Zador, 2013*), numbers (*Bronfman et al., 2015*; *Cisek et al., 2009*), or symbols representing category probabilities (*Yang and Shadlen, 2007*). We refer to this configuration as the DSS paradigm. In this paradigm, the perceptual evidence provided by each sample can be controlled independently, allowing for detailed analyses of how different samples contribute to the behavioral response. The DSS framework can be contrasted with experiments in which the experimenter specifies only the mean stimulus strength on each trial, and variations in sensory evidence over time are not finely controlled or are not easily determined from the raw spatio-temporal stimulus.

Tasks using the DSS paradigm are classically thought to rely on sequential accumulation of the stimulus evidence (*Bogacz et al., 2006*), which we refer to here as temporal integration. *Figure 1A* shows an example stimulus sequence composed of $n$ samples that provide differing amounts of evidence in favor of one alternative vs. another ('A' vs. 'B'). In the temporal integration model, the accumulated evidence fluctuates as new samples are integrated and finishes at a positive value indicating overall evidence for stimulus category A (*Figure 1B*). This integration process can be formalized by defining the decision variable or accumulated evidence $x_i$ and its updating dynamics across stimulus samples: $x_i = x_{i-1} + m_i$ where $m_i = S_i + \varepsilon_i$ represents a noisy version of the true stimulus evidence $S_i$ in the $i$th sample corrupted by sensory noise $\varepsilon_i$. The binary decision $r$ is simply based on the sign of the accumulated evidence $x_n$ at the end of the sample sequence (composed of $n$ samples): $r = A$ if $x_n > 0$, and $r = B$ if $x_n < 0$. This procedure corresponds to the normative strategy with uniform weighting that maximizes accuracy. For such perfect integration, $x_n = \Sigma_i S_i + \Sigma_i \varepsilon_i$, so that the probability of response A is $p(r = A) = \Phi(\Sigma_i \beta S_i)$ where $\Phi$ is the cumulative normal distribution function (the normative weight for the stimuli $\beta$ depends on the noise variance $Var(\varepsilon)$ and the number of samples through $\beta = 1/\sqrt{n \, Var(\varepsilon)}$). Departures from optimality in the accumulation process such as accumulation leak, categorization dynamics, divisive normalization, sensory adaptation, or sticky boundaries may however yield unequal weighting of the different samples (*Yates et al., 2017*; *Brunton et al., 2013*; *Prat-Ortega et al., 2021*; *Bronfman et al., 2016*; *Keung et al., 2019*; *Keung et al., 2020*). To accommodate for these, we allowed the integration model to take any arbitrary weighting of the samples: $p(r = A) = \Phi(\beta_0 + \Sigma_i \beta_i S_i)$ (see Methods for details). The mapping from final accumulated evidence to choice was probabilistic, to account for the effects of noise from different sources in

the decision-making process (*Drugowitsch et al., 2016*). Thus, this model represented an approximate statistical description for any generative model relying on temporal integration of the stimulus evidence.

Although it has been commonly assumed that observers use evidence-integration strategies to perform these psychophysical tasks, recent work has suggested that observers may employ non-integration strategies instead (*Stine et al., 2020*). Here, we consider two specific alternative models. The first non-integration model corresponds to an *extrema-detection* model (*Waskom and Kiani, 2018*; *Stine et al., 2020*; *Ditterich, 2006*). In this model, observers do not integrate evidence across samples but instead base their decision on extreme or salient bits of evidence. More specifically, the observer commits to a decision based on the first sample $i$ in the stimulus sequence that exceeds one of the two symmetrical thresholds, that is such that $|m_i| \geq \theta$. In our example stimulus, the first sample that reaches this threshold in evidence space is the fifth sample, which points toward stimulus category B, so response B is selected (*Figure 1C*). This policy can be viewed as a memoryless decision process with sticky bounds. If the stimulus sequence contains no extreme samples, so that neither threshold is reached, the observer selects a response at random. We also explored an alternative mechanism where in such cases the response is based on the last sample in the sequence, following *Stine et al., 2020*; and a variant of the model where the decision threshold is different on every sample position.

The second non-integration model corresponds to the *snapshot model* (*Stine et al., 2020*; *Pinto et al., 2018*). In this model, the observer attends to only one sample $i$ within the stimulus sequence, and makes a decision based solely on the evidence from the attended sample: $r = A$ if $m_i > 0$, and $r = B$ if $m_i < 0$. The position in the sequence of the attended sample is randomly selected on each trial. In our example, the fourth sample is randomly selected, and since it contains evidence toward stimulus category A, response A is selected (*Figure 1D*). We considered variants of this model that gave it additional flexibility, including: allowing the prior probability over the attended sample to depend on its position in the sequence using a non-parametric probability mass function estimated from the data; allowing for deterministic vs. probabilistic decision-making rule based on the attended evidence; including attentional lapses that were either fixed to 0.02 (split equally between leftward and rightward responses) or estimated from behavioral data. We finally considered a variant of the snapshot model where the decision was made based on a subsequence of $K$ consecutive samples within the main stimulus sequence ($1 \leq K < n$), rather than based on a single sample.

## Standard behavioral statistics favor integration accounts of pulse-based motion perception in primates

To compare the three decision-making models defined above (i.e., temporal integration, extrema-detection, snapshots), we first examined behavioral data from two monkeys performing a fixed-duration motion integration task (*Yates et al., 2017*). In this experiment, each stimulus was composed of a sequence of 7 motion samples of 150 ms each where the motion strength toward left or right was manipulated independently for each sample. At the end of the stimulus sequence, monkeys reported with a saccade whether the overall sequence contained more motion toward the left or right direction. The animals performed 72,137 and 33,416 trials for monkey N and monkey P, respectively, allowing for in-depth dissection of their response patterns.

We fit the three models (and their variants) to the responses for each animal individually (see *Figure 2—figure supplement 1* for estimated parameters for the different models). We then simulated the fitted model and computed, for simulated and experimental data, the psychophysical kernels capturing the weights of the different sensory samples based on their position in the stimulus sequence (*Figure 2B*). Psychophysical kernels were non-monotonic and differed in shape between the two animals, probably reflecting the complex contributions of various dynamics and suboptimalities along the sensory and decision pathways (*Yates et al., 2017*; *Levi and Huk, 2020*).

The temporal profile of the kernel was perfectly matched by the integration model, almost by design, as we gave full flexibility to the model to adjust the sample weights. The snapshot model was provided with similar flexibility, as the prior probability of attending each sample could be fully adjusted to the monkey decisions. However, the snapshot model could not match the experimental psychophysical kernel as accurately. It consistently underestimated the magnitude of weighting in monkey P (*Figure 2B*, bottom row). The extrema-detection model was not endowed with such flexibility of sensory weighting. On the contrary, since the decision was based on the first sample in the

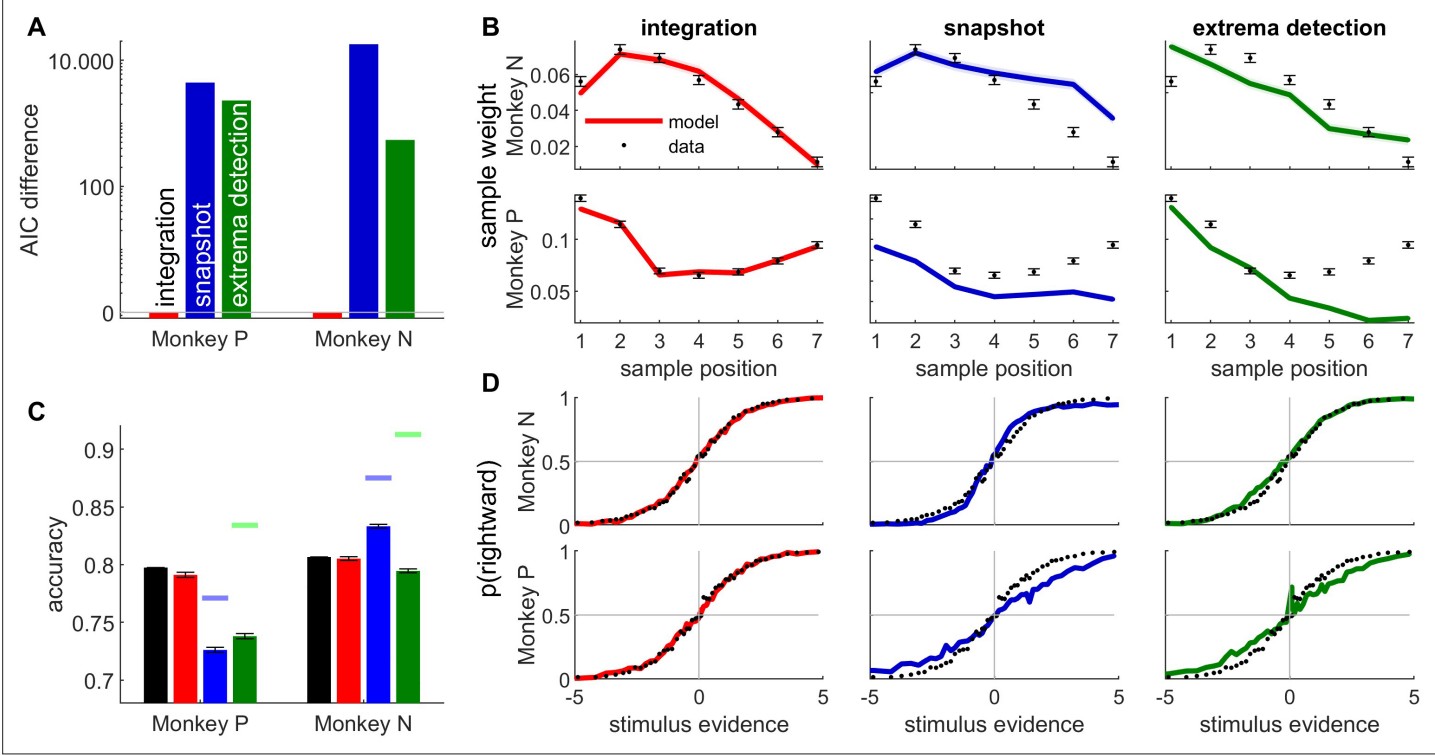

**Figure 2.** The integration model better described monkey behavior than non-integration models. (**A**) Difference between Akaike information criterion (AIC) of models (temporal integration: red bar; snapshot model: blue; extrema-detection model: green) and temporal integration model for each monkey. Positive values indicate poorer fit to data. (**B**) Psychophysical kernels for behavioral data (black dots) vs. simulated data from temporal integration model (left panel, red curve), snapshot model (middle panel, blue curve), and extrema-detection model (right panel, green curve) for the two animals (monkey N: top panels; monkey P: bottom panels). Each data point represents the weight of the motion pulse at the corresponding position on the animal/model response. Error bars and shadowed areas represent the standard error of the weights for animal and simulated data, respectively. (**C**) Accuracy of animal responses (black bars) vs. simulated data from fitted models (color bars), for each monkey. Blue and green marks indicate the maximum performance for the snapshot and extrema-detection models, respectively. Error bars represent standard error of the mean. (**D**) Psychometric curves for animal (black dots) and simulated data (color lines) for monkey N, representing the proportion of rightward choices per quantile of weighted stimulus evidence.

The online version of this article includes the following figure supplement(s) for figure 2:

**Figure supplement 1.** Parameter fits for integration and non-integration models.

**Figure supplement 2.** Model fits for variants of the snapshot model.

**Figure supplement 3.** Model fits for variants of the extrema-detection model.

sequence reaching a certain criterion, this inevitably generates a primacy effect in the psychophysical kernels – or at best a flat weighting (*Stine et al., 2020*). The model thus failed to capture the non-monotonic psychophysical kernels from animal data.

Next, we looked at the psychometric curves and choice accuracy predictions of each fitted model (*Figure 2C, D*). *Stine et al., 2020* have argued that integration and non-integration models can capture the psychometric curves equally well. For both animals, the accuracy and psychometric curves were accurately captured by the integration model. In line with Stine et al., we also found that both non-integration models could reproduce the shape of the psychometric curve in monkey N, although the quantitative fit was always better for the integration than non-integration models. By contrast both non-integration models failed to capture the psychometric curve for monkey P (*Figure 2B*, bottom row). More systematically, the overall accuracy, which is an aggregate measure of the psychometric curve, clearly differed between models, as the accuracy of the non-integration models systematically deviated from animal data for both animals (*Figure 2C*). In other words, all models produce the same type of psychometric curves up to a scaling factor, and this scaling factor (directly linked to the model accuracy) is key to differentiate model fits. For the snapshot model in monkey P, this discrepancy was explained because the model, limited to using one stimulus sample, could not reach the performance

of the animal (compare the maximum accuracy of the model indicated by the blue mark with the accuracy of the animal). (This also explains why the psychophysical kernel of the snapshot model underestimated the true kernel in monkey P.) For the extrema-detection model in monkey P and for both non-integration models in the other animal (monkey N), the model accuracy is not bounded below the subject's accuracy. In such cases, the model can produce better-than-observed accuracy for certain parameter ranges, but these are not the parameters found by the maximum likelihood procedure, probably because they produce a pattern of errors that is inconsistent with the observed pattern of errors. This indicates an inability of the models to match the pattern of errors of the animal (see Discussion).

Finally, we assessed quantitatively which model provided the best fit, while correcting for model complexity using the Akaike information criterion (AIC, *Figure 2A*). In both monkeys, AIC favored the integration model over the two non-integration models by a very large margin. We also explored whether (previously unpublished) elaborations of the extrema-detection and snapshot models could provide a better match to the behavioral metrics considered above (*Figure 2—figure supplements 2 and 3*). We found using the AIC metric that the integration model was preferred over all variants of both non-integration models, for both monkeys. Moreover, these model variants could not replicate the psychophysical kernels as well as the integration model did (*Figure 2—figure supplements 2 and 3*).

In conclusion, while psychometric curves may not always discriminate between integration and non-integration strategies, other metrics including psychophysical kernels, predicted accuracy and quality of fit (AIC) support temporal integration in monkey perceptual decisions. For one model in one monkey (the snapshot model in monkey P), even the simple metric of overall accuracy compellingly supported temporal integration (*Figure 2C*). For the other monkey and/or model, where the distinction was less clear, our model-based approach allowed us to leverage these other metrics to reveal strong support for the temporal integration model (*Figure 2A–C*). While these data rely only on two experimental subjects, we show below further evidence supporting the integration model in humans and rats.

## Temporal integration is more likely than the extrema-detection model: evidence from a unique subset of trials

While formal model comparison leads us to reject the non-integration models in favor of the integration models, it is informative to examine qualitative features of the animal strategies and identify how non-integration models failed to capture them. We started by designing two analyses aimed at testing whether choices were consistent with the extrema-detection model, namely by testing whether choices were strongly correlated with the largest evidence samples. In the first analysis, we looked at the subset of trials where the evidence provided by the largest evidence sample in the sequence was at odds with the total evidence in the sequence: we show one example in *Figure 3B*, where the largest evidence sample points toward response B, while the overall evidence points toward response A. These '*disagree trials*' represent a substantial minority of the whole dataset: 1865 trials (2.6%) in monkey N and 1831 trials (5.5%) in monkey P. If integration is present, the response of the animal should in general be aligned with the total evidence from the sequence (*Figure 3A*, red bars). By contrast, if it followed the extrema-detection model (*Figure 1C*), it should in general follow the largest evidence sample (*Figure 3A*, green bars). In both monkeys, animal choices were more often than not aligned with the integrated evidence (*Figure 3A*, black bars), as predicted by the integration model. The responses generated from the extrema-detection model tended to align more with the largest evidence sample, although that behavior was somehow erratic (for monkey N) due to the large estimated decision noise in the model. This rules out that monkey decisions rely on a memoryless strategy of simply detecting large evidence samples, discarding all information provided by lower evidence samples. Our results complement a previous analysis on disagree trials in this task (*Levi et al., 2018*), by explicitly comparing monkey behavior to model predictions.

We reasoned that the extrema-detection would also leave a clear signature in the 'subjective weight' of the samples, defined as the impact of each sample on the decision as a function of absolute sample evidence (*Yang and Shadlen, 2007*; *Waskom and Kiani, 2018*; *Nienborg and Cumming, 2007*). The extrema-detection model predicts that, in principle, samples whose evidence is below the threshold have little impact on the decision, while samples whose evidence is above the threshold have full

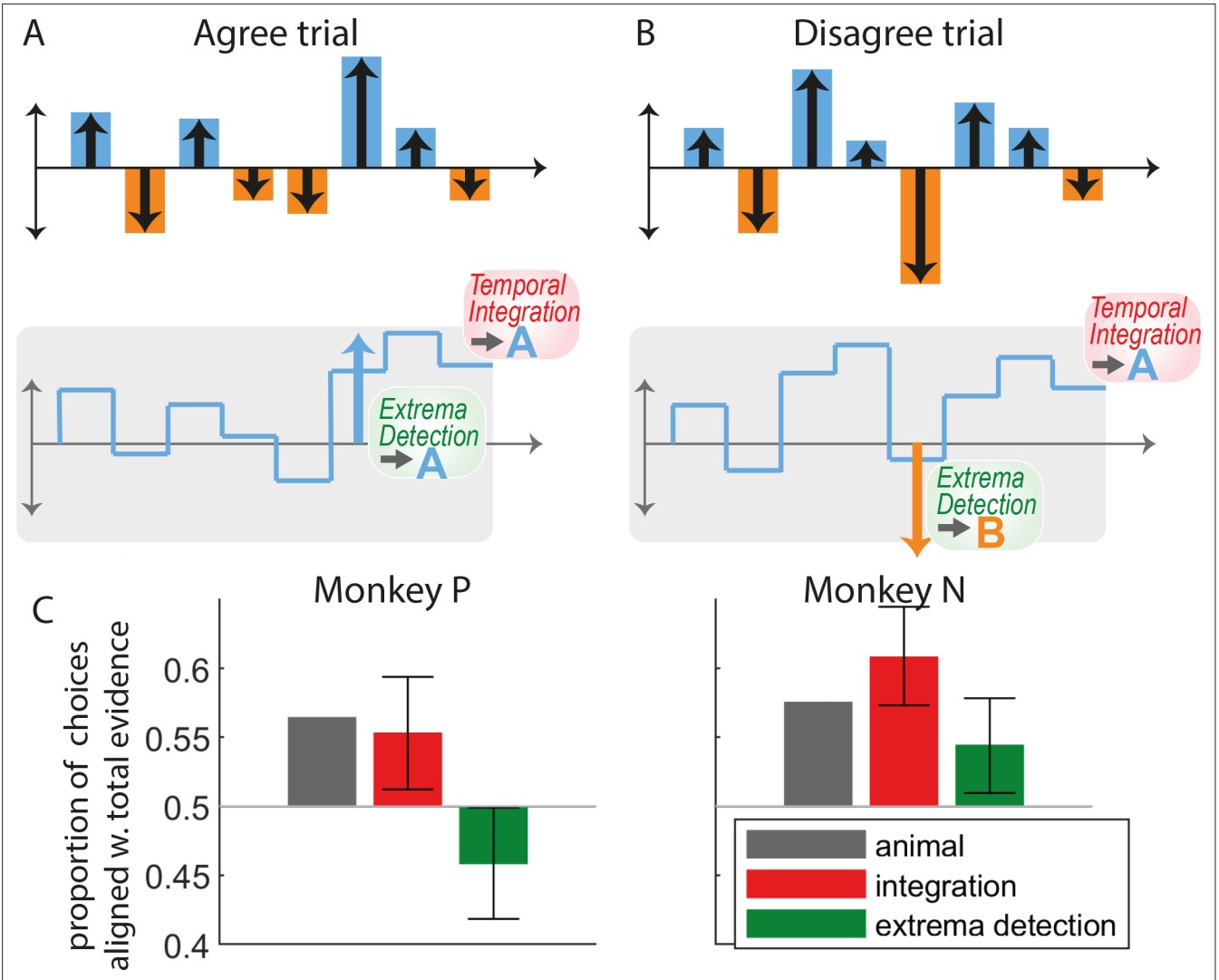

**Figure 3.** The pattern of animal choices is incompatible with extrema-value-based decisions. (**A**) Example of an 'agree trial' where the total stimulus evidence (accumulated over samples) and the evidence from the largest evidence sample point toward the same response (here, response A). In this case, we expect that temporal integration and extrema-detection will produce similar responses (here, A). (**B**) Example of a 'disagree trial', where the total stimulus evidence and evidence from the largest evidence sample point toward opposite responses (here A for the former; B for the latter). In this case, we expect that integration and extrema-detection models will produce opposite responses. (**C**) Proportion of choices out of all *disagree trials* aligned with total evidence, for animal (gray bars), integration (red), and extrema-detection model (green). Error bars denote 95% confidence intervals based on parametric bootstrap (see Methods).

The online version of this article includes the following figure supplement(s) for figure 3:

**Figure supplement 1.** Subjective weights for animal data and simulated models.

impact on the decision. By contrast, the integration model predicts that subjective weight should grow linearly with sample evidence. We estimated subjective weights from monkey choices using a regression method similar in spirit to previous methods (*Yang and Shadlen, 2007*; *Waskom and Kiani, 2018*), taking the form $p\left(r_t = A\right) = \sigma\left(\beta_0 + \Sigma_{i \in [1...n]} \beta_i f\left(S_{t,i}\right)\right)$, where σ is a sigmoidal function. Here *f* is a function that captures the subjective weight of the sample as a function of its associated evidence. Whereas previous methods estimated subjective weights assuming a uniform psychophysical kernel, our method estimated simultaneously subjective weights $f(S)$ and the psychophysical kernel β, thus removing potential estimation biases due to unequal weighting of sample evidence (see Methods). In both monkeys, we indeed found that the subjective weight depends linearly on sample evidence for low to median values of sample evidence (motion pulse lower than 6), in agreement

with the integration model (*Figure 3—figure supplement 1*). Counter to our predictions, simulated data of the extrema-detection model displayed the same linear pattern for low to median values of sample evidence. We realized this was due to the very high estimated sensory noise (*Figure 2—figure supplement 1*), such that, according to the model, even samples with minimal sample evidence were likely to reach the extrema-detection threshold. In other words, unlike the previous analyses, inferring the subjective weights used by animals was inconclusive as to whether animals deployed the extrema-detection strategy. This somewhat surprising dependency reinforces the importance of validating intuitions by fitting and simulating models (*Wilson and Collins, 2019*).

## Impact of early and late stimulus evidence onto choice shows direct evidence for temporal integration

Following model comparisons favoring integration over both snapshot and extrema-detection models, the immediately previous analysis relied on a special subset of trials to provide an additional, and perhaps more intuitive, signature of integration, which ruled out extrema-detection as a possible strategy of either monkey. We next employed another novel analysis specifically designed to tease apart unique signatures of the integration and snapshot models. More specifically, we tested whether decisions were based on the information from only one part of the sequence, as predicted by the snapshot model, or from the full sequence, as predicted by the integration model. To facilitate the analysis, we defined *early evidence* $E_t$ by grouping evidence from the first three samples in the sequence, and *late evidence* $L_t$, as the grouped evidence from the last four samples. We then displayed the proportion of rightward responses as a function of both early and late evidence in a graphical representation that we call *integration map* (*Figure 4A*). A pure integration strategy corresponds to summing early and late evidence equally, which can be formalized as $p(r) = \sigma(E_t + L_t)$, where $\sigma$ is a sigmoidal function. Because this only depends on the sum $E_t + L_t$, the probability of response is invariant to changes in the $(E_t, L_t)$ space along the diagonals, which leave the sum unchanged. These diagonals correspond to isolines of the integration map (*Figure 4A*, left; *Figure 4—figure supplement 2A*). In other words, straight diagonal isolines in the integration map reflect the fact that the decision only depends on the sum of evidence $E_t + L_t$. Straight isolines thus constitute a specific signature of evidence integration.

We contrasted this integration map with the one obtained from a non-integration strategy (*Figure 4A*, middle panel; *Figure 4—figure supplement 2B*). There we assumed that the decision depends either on the early evidence or on the late evidence, as in the snapshot model, with equal probability. This can be formalized as $p(r) = 0.5\sigma(E_t) + 0.5\sigma(L_t)$. In this case, if late evidence is null ($\sigma(L_t) = 0.5$) and early evidence is very strong toward the right ($\sigma(E_t) \simeq 1$) the overall probability for rightward response is $p(r) = 0.75$. This probability contrasts with that obtained in the integration case where the early evidence would dominate and lead to an overwhelming proportion of rightward responses, that is $p(r) \simeq 1$. The 25% of leftwards responses yielded by the non-integration model correspond to trials where only the late (uninformative) part of the stimulus is attended and a random response to the left is drawn. More generally, in regions of the space in which either early or late evidence take large absolute values, their corresponding probability of choice saturates to 0 or 1, when that evidence is attended, so the overall response probability becomes only sensitive to the other evidence. As a result, the equiprobable lines bend toward the horizontal and vertical axes (*Figure 4A*, middle). Finally, to compare predictions from both integration and non-integration models to monkey behavior, we plotted the integration maps for both monkeys (*Figure 4A*, right; *Figure 4—figure supplement 1A*). The isolines were almost straight diagonal lines and showed no consistent curvature toward the horizontal and vertical axes. This provides direct evidence that monkey responses predominantly depend on the sum of early and late evidence – a clear signature of temporal integration.

We derived subsequent tests based on the integration map. We computed conditional psychometric curves as the probability for rightward responses as a function of early evidence $E_t$, conditioned on late evidence value $L_t$ (*Figure 4B*; *Figure 4—figure supplement 1B*). From the integration formula $p(r) = \sigma(E_t + L_t)$, we see that a change in late evidence value corresponds to a horizontal shift of the conditional psychometric curves. By contrast, according to the non-integration formula $p(r) = 0.5\sigma(E_t) + 0.5\sigma(L_t)$, conditioning on different values of late evidence adds a fixed value to the response probability irrespective of early evidence, a vertical shift akin to that introduced by lapse responses (*Figure 4B*, middle panel). The conditional psychometric curves for monkeys (*Figure 4B*,

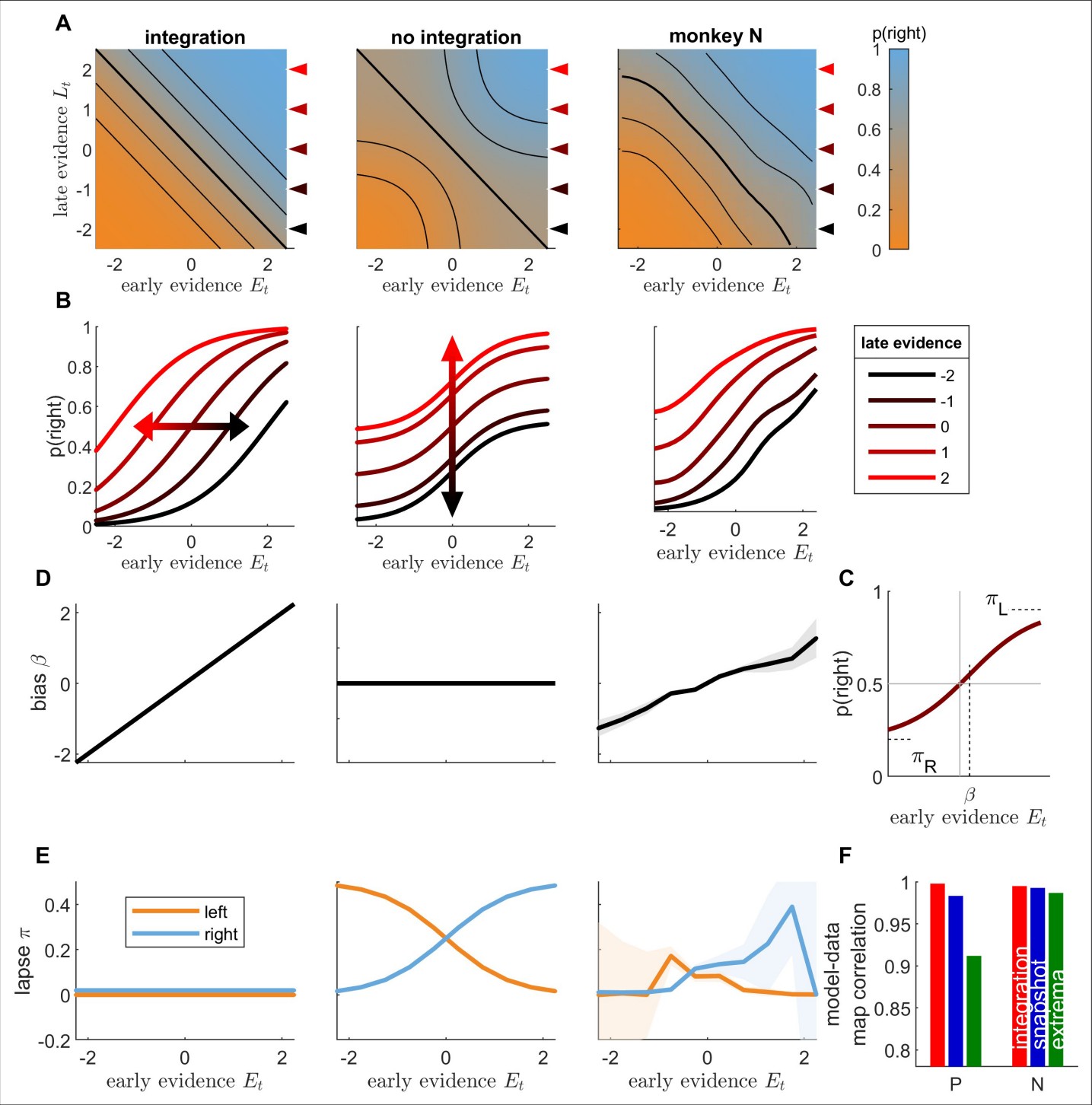

**Figure 4.** Integration of early and late evidence into animal responses is incompatible with the snapshot model. (**A**) Integration map representing the probability of rightward responses (orange: high probability; blue: low probability) as a function of early stimulus evidence $E_t$ and late stimulus evidence $L_t$, illustrated for a toy integration model (where $p\left(right\right) = \sigma\left(E_t + L_t\right)$; left panel) and a toy non-integration model ($p\left(right\right) = 0.5\sigma\left(E_t\right) + 0.5\sigma\left(L_t\right)$; middle panel), and computed for monkey N responses (right panel). Black lines represent the isolines for p(rightwards) = 0.15, 0.3, 0.5, 0.7, and 0.85. (**B**) Conditional psychometric curves representing the probability for rightward response as a function of early evidence $E_t$, for different values of late evidence $L_t$ (see inset for $L_t$ values), for toy models and monkey N. The curves correspond to horizontal cuts in the integration maps at $L_t$ values marked by color triangles in panel A. (**C**) Illustration of the fits to conditional psychometric curves. The value of the bias $\beta$, left lapse $\pi_L$ and right lapse $\pi_R$ are estimated from the conditional psychometric curves for each value of late evidence. (**D**) Lateral bias as a function of late evidence for toy models and monkey N. Shaded areas represent standard error of weights for animal data. (**E**) Lapse parameters (blue: left lapse;

*Figure 4 continued on next page*

*Figure 4 continued*

orange: right lapse) as a function of late evidence for toy models and monkey N. (**F**) Pearson correlation between integration maps for animal data and integration maps for simulated data, for each animal. Red: integration model; blue: snapshot model; green: extrema-detection model.

The online version of this article includes the following figure supplement(s) for figure 4:

**Figure supplement 1.** Integration of early and late evidence for monkey P.

**Figure supplement 2.** Integration between early and late evidence for simulated data from integration and non-integration models.

**Figure supplement 3.** Individual Lateral Intra Parietal (LIP) neurons integrate sensory information over stimulus sequence.

right panel; *Figure 4—figure supplement 1* and *Figure 4—figure supplement 2*) displayed horizontal shifts as late evidence was changed, consistently with the integration hypothesis. We sought to quantify these shifts in better detail. To this purpose, we fitted each conditional psychometric curve with the formula $p\left(r\right) = \left(1 - \pi_L - \pi_R\right)\sigma\left(\alpha E_t + \beta\right) + \pi_R$, where $\pi_L$, $\pi_R$, $\alpha$, and $\beta$ correspond to the left lapse, right lapse, sensitivity, and lateral bias parameters, respectively (*Figure 4C*, *Figure 4—figure supplement 1* and *Figure 4—figure supplement 2*). The integration model predicts that the bias parameter $\beta$ should vary linearly with $L_t$, while lapse parameters should remain null (*Figure 4D*, left panel). By contrast, the non-integration model predicts that the horizontal shift parameter $\beta$ should remain constant while left and right lapse parameters $\left(\pi_L, \pi_R\right)$ should vary (middle panel), as these lapse parameters correspond to the trials where early evidence is not attended and the response depends simply on late evidence. Both monkeys showed a very strong linear dependence between late evidence and the horizontal shift $\beta$ (*Figure 4D*, right panel; see also *Figure 4—figure supplement 1*), further supporting that late evidence is summed to early evidence. By contrast, the lapse parameters showed no consistent relationship with late evidence $L_t$ (*Figure 4E*, right panel). Finally, we directly assessed the similarities between the integration maps from monkey responses and from simulated responses for the three models (integration, snapshot, and extrema-detection). The model-data correlation was larger in the integration model than in the non-integration strategies for both monkeys (*Figure 4E*; unpaired *t*-test on bootstrapped *r* values: p < 0.001 for each animal and comparison against extrema-detection and against snapshot model). Overall, integration maps allow to dissect how early and late parts of the stimulus sequence are combined to produce a behavioral response. In both monkeys, these maps carried signatures of temporal integration. For monkey N, the integration model and the data look very similar (*Figure 4—figure supplement 2*). For monkey P, there is still a qualitative dependency that deviates from non-integration, but which is not as uniquely matched to the integration strategy (although the imperfect coverage of the two-dimensional space impedes further investigations; *Figure 4—figure supplement 1*). Thus, complementing the statistical model tests favoring integration, this richer visualization allows the data to show us that some degree of integration is occurring, albeit not perfect.

## Visual orientation discrimination in humans relies on temporal integration

Overall, all our analyses converged to support the idea that monkey decisions in a fixed-duration motion discrimination task relied on temporal integration. We explored whether the same results would hold for two other species and perceptual paradigms. We first analyzed the behavioral responses from nine human subjects performing a variable-duration orientation discrimination task (*Cheadle et al., 2014*). In each trial, a sequence of 5–10 gratings with a certain orientation were shown to the subject, and the subject had to report whether they thought the gratings were overall mostly aligned to the left or to the right diagonal. In this task, the experimenter can control the evidence provided by each sample by adjusting the orientation of the grating. We performed the same analyses on the participant responses as on monkey data. As for monkeys, we found that the integration model nicely captured psychometric curves, participant accuracy and psychophysical kernels (*Figure 5A–C*, red curves and symbols). By contrast, both non-integration models failed to capture these patterns (*Figure 5A–C*, blue and green curves and symbols). The accuracy from both models consistently underestimated participant performance: eight and six out of nine subjects outperformed the maximum performance for the snapshot and extrema-detection models, respectively (*Figure 5—figure supplement 1*). This suggests that human participants achieved such accuracy by integrating sensory evidence over successive samples. Moreover, subjects overall weighted more later samples

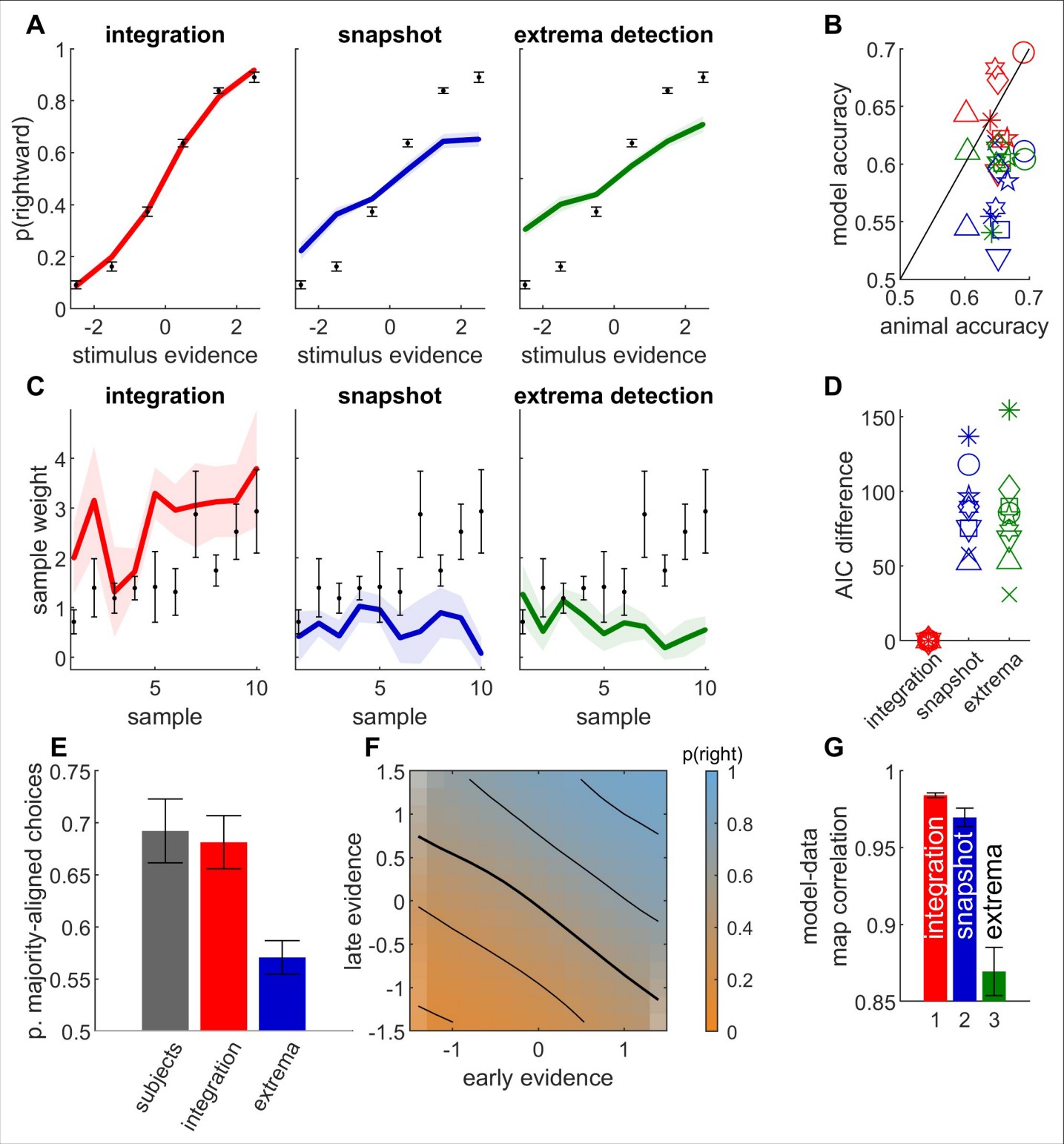

**Figure 5.** Behavioral data from orientation discrimination task in humans provide further evidence for temporal integration. (**A**) Psychometric curves for human and simulated data, averaged across participants (*n* = 9). Legend as in *Figure 2C*. (**B**) Simulated model accuracy (*y*-axis) vs. participant accuracy (*x*-axis) for integration model (red), snapshot model (blue) and extrema-detection model (green). Each symbol corresponds to a participant. (**C**) Psychophysical kernel for human and simulated data, averaged across participants. Legend as in A. (**D**) Difference in Akaike information criterion (AIC) between each model and the integration model. Legend as in B. (**E**) Proportion of choices aligned with total stimulus evidence in disagree trials, for participant data (gray bars) and simulated models, averaged over participants. (**F**) Integration map for early and late stimulus evidence, computed as

*Figure 5 continued on next page*

*Figure 5 continued*

in *Figure 4A*, averaged across participants. (**G**) Correlation between integration map of participants and simulated data for integration, snapshot, and extrema-detection models, averaged across participants. Color code as in B. Error bars represent the standard error of the mean across participants in all panels.

The online version of this article includes the following figure supplement(s) for figure 5:

**Figure supplement 1.** Maximum accuracy of the non-integration models vs. human subject accuracy in the orientation discrimination task.

(*Figure 5C*), which is inconsistent with the extrema-detection mechanism. A formal model comparison confirmed that in each participant, the integration model provided a far better account of subject responses than either of the non-integration models did (*Figure 5D*). We then assessed how subjects combined information from weak and strong evidence samples into their decisions, using the same analyses as for monkeys. As predicted by the integration model, but not by the extrema-detection model, human choices consistently aligned with the total stimulus evidence and not simply with the strongest evidence sample (*Figure 5E*). Finally, the average integration map for early and late evidence within the stimulus sequence displayed nearly linear diagonal isolines, showing that both were integrated into the response (*Figure 5F*). Integration maps from participants correlated better with maps predicted by the integration model than with maps predicted by either of the alternative non-integration strategies (*Figure 5G*; two-tailed *t*-test on bootstrapped *r* values: p < 0.001 for six out of nine participants in the integration vs. snapshot comparison; in all nine participants for the integration vs. extrema-detection comparison). Overall, these analyses provide converging evidence that human decisions in an orientation discrimination task rely on temporal integration.

## Auditory intensity discrimination in rats relies on temporal integration

Finally, we analyzed data from five rats performing a fixed-duration auditory task where the animals had to discriminate the side with larger acoustic intensity (*Pardo-Vazquez et al., 2019*). The relative intensity of the left and right acoustic signals was modulated in sensory samples of 50ms, so that the stimulus sequence provided time-varying evidence for the rewarded port. The stimulus sequence was composed of either 10 or 20 acoustic samples of 50 ms each, for a total duration of 500 or 1000 ms. We applied the same analysis pipeline as for monkey and human data. The integration model provided a much better account of rat choices than non-integration strategies, based on psychometric curves (*Figure 6A*), predicted accuracy (*Figure 6B*), psychophysical kernel (*Figure 6C*), and model comparison using AIC (*Figure 6D*). Similar to humans and monkeys, rats tended to select the side corresponding to the total stimulus evidence and not the largest sample evidence in 'disagree' trials, as predicted by the integration model (*Figure 6E*). Finally, the integration map was largely consistent with an integration strategy (*Figure 6F*), and correlated more strongly with simulated maps from the integration model (unpaired *t*-test on bootstrapped *r* values: p < 0.001 for each animal and comparison against extrema-detection and against snapshot model).

## Discussion

We investigated the presence of temporal integration in perceptual decisions in monkeys, humans, and rats through a series of standard and innovative analyses of response patterns. In all analyses, we contrasted predictions from one integration and two non-integration computational models of behavioral responses (*Figure 1*). For each non-integration model, we considered multiple variants to explore the maximal flexibility offered by each framework to capture animal behavior. For our datasets, evidence in favor of integration was easy to achieve using standard model comparison techniques as well as comparing simulated psychometric curves and psychophysical kernels to their experimental counterparts (*Figure 2*). Our results are in line with previous evidence for temporal integration in perceptual decisions of humans and mice (*Pinto et al., 2018*; *Stine et al., 2020*; *Waskom and Kiani, 2018*). Importantly, we also put forth new analyses targeted at revealing specific signatures of temporal integration.

   In some cases, we could link the failure of the non-integration model to a fundamental limitation of the model. For example, the extrema-detection model cannot explain the non-monotonic psychophysical kernels of monkeys or the increasing psychophysical kernels in humans. This is because the decision in that mode is based on the first sample to reach a certain fixed criterion, so it will always

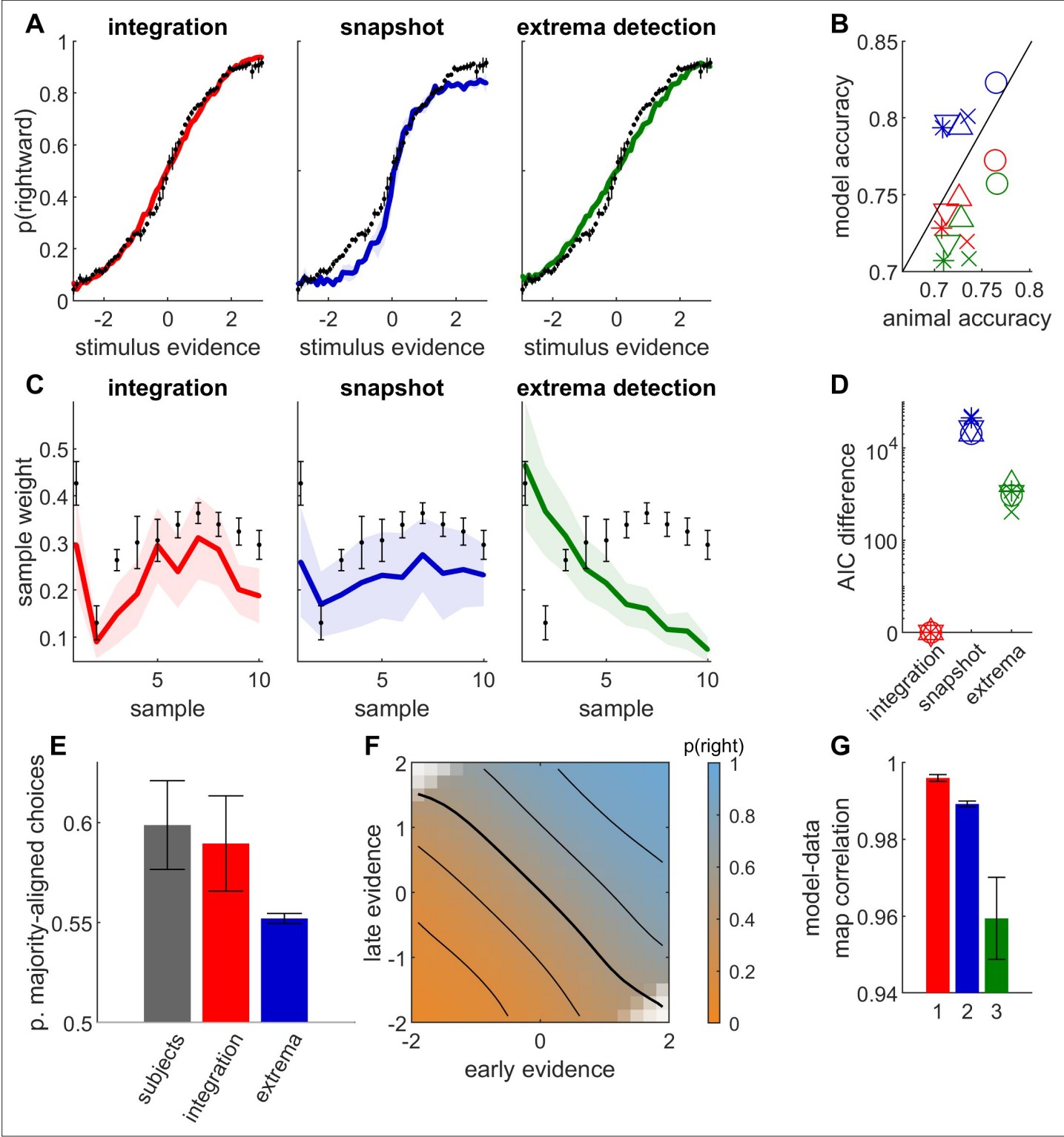

**Figure 6.** Behavioral data from auditory discrimination task in five rats provide further evidence for temporal integration. (**A-G**) Legend as in *Figure 5*. Rats were rewarded for correctly identifying the auditory sequence of larger intensity (number of samples: 10 or 20; stimulus duration: 500 or 1000 ms). Legend as in *Figure 5*. Psychophysical kernels are computed only for 10-sample stimuli (in 4 animals). See *Figure 6—figure supplement 1* for psychophysical kernels with 20-sample stimuli.

The online version of this article includes the following figure supplement(s) for figure 6:

**Figure supplement 1.** Psychophysical kernels for animals and models in rats (*n* = 3) performing the discrete-sample stimulus (DSS) task with 20-sample stimuli.

produce a primacy effect, that is, a decreasing psychophysical kernel. Although this effect can be small, and in practice yields approximately flat kernels (*Stine et al., 2020*), it cannot produce increasing or non-monotonic kernels.

Another strong limitation of non-integration models (both the extrema-detection and the snapshot model) is that accuracy is limited by the fact that decisions depend on a single sample. We found that that boundary performance (i.e., the maximum performance that a model can reach) was actually lower than subject accuracy for most human participants, de facto ruling out these non-integration strategies for these participants. This is consistent to what was observed in a contrast discrimination DSS task where human subjects had to make judgments about image sequences spanning up to tens of seconds each (*Waskom and Kiani, 2018*). It clearly contrasts however with results from *Stine et al., 2020* where the non-integration strategies matched the accuracy of human subjects performing the classical random-dot-motion task. This discrepancy may be related to the different sources of noise in the two paradigms. In DSS tasks, because the sensory evidence provided by the stimulus at each moment is controlled by the experimenter, the unpredictability of human responses essentially stems from internal noise at the level of sensory processing and temporal integration (*Waskom and Kiani, 2018*; *Drugowitsch et al., 2016*). By contrast, the random dot motion task (*Kiani et al., 2008*), which is a non-DSS task because the experimenter does not typically specify differing amounts of motion in each time epoch within a single trial, typically elicits more variable responses due to the presence of stimulus noise. This overall increased noise level leads to a looser relationship between the stimulus condition and the behavioral responses, which can thus be accounted for by a larger spectrum of computational mechanisms. These issues have been addressed by forcing 'pulses' of a certain stimulus strength and/or by performing post hoc analyses to estimate signal and noise (*Kiani et al., 2008*; *Huk and Shadlen, 2005*) but these are partial solutions that DSS paradigms solve by design. This illustrates the benefits of using experimental designs where variability in stimulus information can be fully controlled and parametrized by the experimenter, as these paradigms discriminate more precisely between different models of perceptual decisions (see *Cisek et al., 2009*).

In at least one monkey, although quantitative metrics such as penalized log-likelihood and fits to psychometric curves clearly pointed to the integration model as the best account to behavior, the qualitative failure modes of the non-integration strategies (especially the snapshot model) was not immediately clear. Although we tried variants for each non-integration model, there remained a possibility that our precise implementation failed to account for monkey behavior but that other possible implementations would. Note that the extrema-detection and snapshot are two of the many possible non-integration strategies. A generic form for non-integration strategies corresponds to a policy that implements position-dependent thresholds on the instantaneous sensory evidence. In this framework, the extrema-dependent model corresponds to the case with a position-independent threshold, while the snapshot model corresponds to a null bound for one sample and infinite bounds for all other samples. To rule out these more complex strategies, we conducted additional analyses that specifically targeted core assumptions of the integration and non-integration strategies.

First, the extrema-detection model fails to account for the data because it predicts that largest evidence samples should have a disproportionate impact on choices. However, this does not occur, as monkeys and humans tend to respond according to the total evidence and not the single large-evidence sample (*Figures 3C and 5E*) – and see *Levi et al., 2018* for a similar analysis. All non-integration strategies share the property that on each trial the decision should only rely either on the early or the late part of the trial. We thus directly examined the assumptions of integration and non-integration models by assessing how the evidence from the early and late parts of each stimulus sequence is combined to produce a decision. We introduced *integration maps* (*Figure 4*) to inspect such integration: isolines of the integration maps will be rectilinear if and only if early and late evidence are summed, in other words if and only if temporal integration takes place. Unequal weighting of evidence would still produce rectilinear isolines, albeit with a different angle. By contrast, a non-integration scenario when on each trial only a single piece of evidence contributes to the decision predicts isolines that bend toward the axes. Integration maps from monkey, human, and rat subjects nicely matched the predictions of the integration models, proving that their decisions do rely on temporal integration. Note that this innovative analysis technique could be used to probe integration of evidence not only at temporal level but also between different sources of evidence. Indeed, there has been an intense debate about whether sensory information from different spatial locations

or different modalities are integrated prior to reaching a decision, or whether decisions are taken separately for each source before being merged, which can be viewed as extensions to the snapshot model (*Pannunzi et al., 2015*; *Otto and Mamassian, 2012*; *Lorteije et al., 2015*; *Hyafil and Moreno-Bote, 2017*). Our integration analysis could provide new answers to this old debate.

Integration maps can be computed not only for choice patterns but for any type of behavioral or neural marker of cognition. We computed a neural integration map (*Figure 4—figure supplement 3*) by looking at the average spike activity of Lateral Intra Parietal (LIP) neurons as a function of early and late evidence, for neurons recorded while the monkeys performed the motion discrimination experiment (*Yates et al., 2017*). The neural integration map clearly showed rectilinear isolines, as predicted by an integration model of neural spiking. By contrast, neural implementations of the snapshot and extrema-detection predicted strongly curved isolines. The activity of LIP neurons correlates with the evidence accumulated over the presentation of the stimulus in favor of either possible choices (*Gold and Shadlen, 2007*). This result shows that the activity of individual LIP neurons indeed reflects the temporal integration of sensory information that drives animal behavior.

We have focused in this study on paradigms where the stimulus duration is fixed by the experimenter, and subjects could only respond after stimulus extinction. Stine et al. proposed a method for distinguishing integration from non-integration strategies but the method is based on two experimental conditions: experiments where stimulus duration is controlled by the experimenter and experiments where the stimulus is controlled by the subject (i.e., plays until the subject responds, 'reaction time paradigms'). Our study puts forth a method to differentiate integration from non-integration strategies that is based on only one experimental condition (i.e., the variable-duration paradigm), and may therefore be applied to existing datasets.

Other studies have shown how integration and non-integration strategies can be disentangled in free reaction time task paradigms. Specifically, different models make different predictions regarding how the total sample evidence presented before response time should vary with response time (*Glickman and Usher, 2019*; *Zuo and Diamond, 2019*). Glickman and Usher used these predictions to rule out non-integration strategies in a counting task in humans, and *Zuo and Diamond, 2019* found evidence for evidence integration to bound when rats discriminate textures using whisker touches. Under strong urgency constraints, it has been proposed that decisions depend on very limited temporal integration of the stimulus by low-pass filtering of stimulus evidence (*Cisek et al., 2009*; *Thura et al., 2012*). However, this suggestion cannot explain the fact that evidence presented early in the trial influences decisions taken later on *Winkel et al., 2014*; *Thura et al., 2012*. Of note, the absence of integration seems a more viable strategy when the duration of the stimulus is controlled externally and the benefits of integrating in terms of accuracy might not compensate for its cognitive cost. In free reaction time paradigms, waiting for a long sequence of samples and selecting its response based on a single sample does not seem a particularly efficient strategy. If the cognitive cost of integration is high, it is more beneficial to interrupt the stimulus sequence early with a rapid response. Indeed, fast or very fast (under 250 ms) perceptual decisions are very common (*Uchida et al., 2006*; *Zariwala et al., 2013*; *Stanford and Salinas, 2021*; *Pardo-Vazquez et al., 2019*; *Hermoso-Mendizabal et al., 2020*). Such rapid termination of the decision process have been attributed either to urgency signals modulating the integration of stimulus evidence (*Drugowitsch et al., 2012*; *Stanford and Salinas, 2021*) or to action initiation mechanisms that time the response after a specific time (e.g., one or two samples) following stimulus onset (*Hernández-Navarro et al., 2021*). Here, we have shown that even in paradigms where the stimulus duration is controlled by the experimenter, mammals often integrate sensory evidence over the entire stimulus.

In conclusion, we have found strong evidence for temporal integration in perceptual tasks across species (monkeys, humans, and rats) and perceptual domain (visual motion, visual orientation, and auditory discrimination). Although the timescale of integration can be adapted to the statistics of the environment (*Ossmy et al., 2013*; *Glaze et al., 2015*; *Kilpatrick et al., 2019*), the principle that stimulus evidence is integrated over time appears to be a hallmark of perception. This evidence was gathered by leveraging experimentally controlled sensory evidence at each sensory sample composing a stimulus, and novel model-based statistical analysis. We speculate that temporal integration is a ubiquitous feature of perceptual decisions due to hard-wired neural integrating circuits, such as recurrent stabilizing connectivity in sensory and perceptual areas (*Wang, 2008*; *Wimmer et al., 2015*).

# Methods

## Monkey experiment

We present here the most relevant features of the behavioral protocol – see *Yates et al., 2017* for further experimental details. Two adult rhesus monkeys performed a motion discrimination task. On each trial, a stimulus consisting of a hexagonal grid (5–7 degrees, scaled by eccentricity) of Gabor patches (0.9 cycle per degree; temporal frequency 5 Hz for Monkey P; 7 Hz for Monkey N) was presented. Monkeys were trained to report the net direction of motion in a field of drifting and flickering Gabor elements with an eye movement to one of two targets. Each trial motion stimulus consisted of seven consecutive motion pulses, each lasting 9 or 10 video samples (150 or 166 ms; pulse duration did not vary within a session), with no interruptions or gaps between the pulses. The strength and direction of each pulse $S_{ti}$ for trial $t$ and sample $i$ were set by a draw from a Gaussian rounded to the nearest integer value. The difficulty of each trial was modulated by manipulating the mean and variance of the Gaussian distribution. Monkeys were rewarded based on the empirical stimulus and not on the stimulus distribution. We analyzed a total of 112 sessions for monkey N and 60 sessions for monkey P, with a total of 72,137 and 33,416 valid trials, respectively. These sessions correspond to sessions with electrophysiological recordings reported in *Yates et al., 2017* and purely behavioral sessions.

## Human experiment

Nine adult subjects performed an orientation discrimination task whereby on each trial they reported in each trial whether a series of gratings were perceived to be mostly tilted clockwise or counter-clockwise (*Drugowitsch et al., 2016*). Each DSS consisted of 5–10 gratings. Each grating was a high-contrast Gabor patch (color: blue or purple; spatial frequency = 2 cycles per degree; SD of Gaussian envelope = 1 degree) presented within a circular aperture (4 degrees) against a uniform gray background. Each grating was presented during 100 ms, and the interval between gratings was fixed to 300 ms. The angles of the gratings were sampled from a von Mises distribution centered on the reference angle ($\alpha_0 = 45$ degrees for clockwise sequences, 135 degrees for anticlockwise sequences) and with a concentration coefficient $\kappa = 0.3$. The normative evidence provided by sample $i$ in trial $t$ in favor of the clockwise category corresponds to how well the grating orientation $\alpha_{ti}$ aligns with the reference orientation, that is $S_{ti} = 2\kappa \, cos \left( 2 \left( \alpha_{ti} - \alpha_0 \right) \right)$.

Each sequence was preceded by a rectangle flashed twice during 100 ms (the interval between the flashes and between the second flash and the first grating varied between 300 and 400 ms). The participants indicated their choice with a button press after the onset of a centrally occurring dot that succeeded the rectangle mask and were made with a button press with the right hand. Failure to provide a response within 1000 ms after central dot onset was classified as invalid trial. Auditory feedback was provided 250 ms after participant response (at latest 1100 ms after end of stimulus sequence). It consisted of an ascending tone (400/800 Hz; 83/167 ms) for correct responses; descending tone (400/400 Hz; 83/167 ms) for incorrect responses; a low tone (400 Hz; 250 ms) for invalid trials.

Trials were separated by a blank interstimulus interval of 1200–1600 ms (truncated exponential distribution of mean 1333 ms). Experiments consisted of 480 trials in 10 blocks of 48. It was preceded with two blocks of initiation with 36 trials each. In the first initiation block, there was only one grating in the sequence, and it was perfectly aligned with one of the reference angles. In the second initiation block, sequences of gratings were introduced, and the difficulty was gradually increased (the distribution concentration linearly decreased from $\kappa = 1.2$ to $\kappa = 0.3$). Invalid trials (mean 6.9 per participant, std 9.4) were excluded from all regression analyses. The study was approved by the local ethics committee (approval 2013/5435/I from CEIm-Parc de Salut MAR).

## Rat experiment

Rat experiments were approved by the local ethics committee of the University of Barcelona (Comité d'Experimentació Animal, Barcelona, Spain, protocol number Ref 390/14). Five male Long-Evans rats (350–650 g), pair-housed and kept on stable conditions of temperature (23°C) and humidity (60%) with a constant light–dark cycle (12:12 hr, experiments were conducted during the light phase). Rats had free access to food, but water was restricted to behavioral sessions. Free water during a limited period was provided on days with no experimental sessions.

Rats performed a fixed-duration auditory discrimination task where they had to classify noisy stimuli based on the intensity difference between the two lateral speakers (*Pardo-Vazquez et al., 2019*; *Hermoso-Mendizabal et al., 2020*). An LED on the center port indicated that the rat could start the trial by poking in that center port. After this poke, rats had to hold their snouts in the central port during 300 ms (i.e., fixation). Following this period, an acoustic DSS was played. Rats had to remain in the central port during the entire presentation of the stimulus. At stimulus offset, the center LED went off and rats could then come out of the center port and head toward one of the two lateral ports. Entering the lateral port associated with the speaker that generated the larger sound intensity led to a reward of 24 µl of water (correct responses), while entering the opposite port lead to a 5-s timeout accompanied with a bright light during the entire period (incorrect responses). If rats broke fixation during the pre-stimulus fixation period or during the stimulus presentation, the sound was interrupted, the center LED remained on, and the rat had to initiate a new trial starting by center fixation followed by a new stimulus. Fixation breaks were not included in any of the analyses. Stimulus duration was 0.5 s (10 samples) or 1 s (20 samples). Two rats performed 0.5-s stimuli only (77,810 and 54,803 valid trials, respectively); one rat performed 1-s stimuli only (42,474 valid trials); the remaining two rats performed a mixture of 0.5 and 1 s stimuli trials randomly interleaved (5016 trials and 65,212 valid trials, respectively, for one animal; 7374 and 38,829 trials for the other animal). In each trial $k$ one stimulus $S_K^X(t)$ was played in each speaker ($X$ = R for the Right speaker and $X$ = L for the Left speaker). Each stimulus was an amplitude-modulated (AM) broadband noise defined by $S_k^X(t) = \left[1 + sin\left(f_{AM}t + \varphi\right)\right] a_k^X(t) \, \xi_X(t)$ where $f_{AM}$ = 20 Hz (sensory samples lasted 50 ms), the phase delay $\varphi = 3\pi/2$ and $\xi_X(t)$ were broadband noise bursts. The amplitudes of each sound in each frame were $a_k^L(t) = \left(1 + S_{k,f}\right)/2$ and $a_k^R(t) = \left(1 - S_{k,f}\right)/2$ with $S_{k,f}(t)$ being the instantaneous evidence that was drawn independently in each frame $f$ from a transformed Beta distribution with support [−1,1]. With this parametrization of the two sounds the sum of the two envelopes was constant in all frames $a_k^L(t) + a_k^R(t)$ = 1. There were 7 × 5 stimulus conditions, each defined by a Beta distribution, spanning 7 mean values (−1, −0.5, −0.15, 0, 0.15, 0.5, and 1) and 5 different standard deviations (0, 0.11, 0.25, 0.57, and 0.8). In around the first half of the sessions, only sample sequences in which the total stimulus evidence matched the targeted nominal evidence were used. This effectively introduced weak correlations between samples. In the second half of the sessions, this condition was removed and samples in each stimulus were drawn independently from the corresponding Beta distribution.

## Integration model

The integration model for human participants corresponds to a logistic regression model, where the probability of selecting the right choice $p\left(r_t\right)$ at trial $t$ depends on the weighted sum of the sample evidence: $p\left(r_t\right) = \sigma\left(\beta_0 + \Sigma_{i \in \left[1...n\right]} \beta_i S_{ti}\right)$, where $\beta_0$ is a lateral bias, $S_{ti}$ is the signed sample evidence at sample $i$; $\beta_i$ is the sensory weight associated with the $i$th sample in the stimulus sequence; and $\sigma\left(x\right) = \left(1 + e^{-x}\right)^{-1}$ is the logistic function. The vector $\beta_i$'s allowed to capture different shapes of psychophysical kernels (e.g., primacy effects, recency effects) which can emerge due to a variety of suboptimalities in the integration process (leak, attractor dynamics, sticky bounds, sensory after-effects, etc.) (*Brunton et al., 2013*; *Yates et al., 2017*; *Prat-Ortega et al., 2021*; *Bronfman et al., 2016*; *Keung et al., 2019*; *Keung et al., 2020*). Technically, our statistical approach corresponds to the first-order expansion of the Volterra series used to approximate any integration model (*Neri, 2004*) by a simple and fittable logistic model. In other words, because a more complete description of integration models would be computationally challenging to fit and prone to overfitting, we chose a statistical approximation in the form of the logistic model that captures the essence of any generative model that include temporal integration, that is weighted summation of stimulus evidence.

For the monkey and rat data, we included a session-dependent modulation gain $\gamma_t$ to capture the large variations in performance in monkeys across the course of sessions (see *Figure 2—figure supplement 1A*):

$$p\left(r_t\right) = \sigma\left(\beta_0 + \gamma_t \Sigma_{i \in \left[1...n\right]} \beta_i S_{ti}\right)$$

This model corresponds to a bilinear logistic regression model which pertains to the larger family of Generalized Unrestricted Models (GUMs) (*Adam and Hyafil, 2020*). Parameters ($\beta, \gamma$) were fitted

using the Laplace approximation as described in *Adam and Hyafil, 2020*. The modulation gain was omitted when applied to human data, yielding a classical logistic regression model.

## Snapshot model

In the snapshot model, decisions are based on each trial based upon a single sample. The model also includes the possibility for left and right lapses. In each trial, the attended sample is drawn from a multinomial distribution of parameters $(\pi_1, ...\pi_n, \pi_L, \pi_R)$, where the first terms $\pi_i$ $(1 \leq i \leq n)$ correspond to the probability of attending sample $i$, and $\pi_L$ and $\pi_R$ correspond to the probability of left and right lapses, respectively. Upon selecting sample $i$, the probability for selecting the right choice is given by the function $H_i(S_t)$. In the deterministic version of the model, $H_i$ is simply determined by the sign of the $i$th sample evidence: $H_i(S_t) = 1$ if $S_{ti} > 0$, $H_i(S_t) = 0$ if $S_{ti} < 0$, and $H_i(S_t) = 0.5$ if $S_{ti} = 0$ (i.e., random guess if the sample has null evidence). We also define similar functions for lapse responses: $H_R(S) = 1$ and $H_L(S) = 0$, irrespective of the stimulus. In the non-deterministic version of the model, the probability $H_i(S_{ti})$ is determined by a logistic function of the attended sample evidence $H_i(S_t) = \sigma(\beta_i S_{ti})$ where $\beta_i$ describes a sensitivity parameter. The deterministic case can be viewed as the limit of the non-deterministic case when all sensitivity parameters $\beta_i$ diverge to $+\infty$, that is when sensory and decision noise are negligible.

The overall probability for selecting right choice (marginalizing over the attended sample, which is a hidden variable) can be captured by a mixture model:

$$p(r_t) = \pi_R + \Sigma_{i \in [1...n]} \pi_i H_i(S_t) = \Sigma_{i \in [1...n,L,R]} \pi_i H_i(S_t)$$

The mixture coefficients $\pi_i$ $(i = 1, ...n, L, R)$ are constrained to be non-negative and sum up to 1. In the non-deterministic model, the parameters also include sensitivity parameters $\beta_i$. The model is fitted using Expectation-Maximization (*Bishop, 2006*). In the Expectation step, we compute the responsibility $z_{ti}$, that is the posterior probability that the sample $i$ was attended at trial $t$ (for $i = L, R$, the probability that the trial corresponded to a lapse trial):

$$z_{ti} = \pi_i \theta(S_{ti}) / \Sigma_j \pi_j H(S_{tj}) \text{ for rightward responses } (R_t = 1)$$

$$z_{ti} = \pi_i (1 - H(S_{ti})) / \Sigma_j \pi_j (1 - H(S_{tj})) \text{ for leftward responses } (R_t = 0)$$

In the Maximization step, we update the value of the parameters by maximizing the Expected Complete Log-Likelihood (ECLL): $Q(\pi, \beta) = \Sigma_{ti} z_{ti} log\, p(r_t; \pi, \beta)$. Maximizing over the mixture coefficients with the unity-sum constraint provides the classical update: $\pi_i = \Sigma_{ti} z_{ti} / N$, where $N$ is the total number of trials. In the non-deterministic model, maximizing the ECLL over sensitivity parameters is equivalent to fitting a logistic regression model with weighted coefficients $z_{ti}$, which is a convex problem. Best fitting parameters can be found using Newton–Raphson updates on the parameters:

$$\beta_i^{(new)} = \beta_i - \frac{\partial Q/\partial \beta_i}{\partial^2 Q/\partial \beta_i^2} \text{ with}$$

$$\partial Q/\partial \beta_i = \sum_t z_{ti}(p(r_t - R_t)) \text{ and } \partial^2 Q/\partial \beta_i^2 = \Sigma_t z_{ti} S_{ti}^2 p(r_t)(1 - p(r_t))$$

To speed up the computations, in each M step, we only performed one Newton–Raphson update for each sensitivity parameter, rather than iterating the updates fully until convergence. The EM procedure was run until convergence, assessed by an increment in the log-likelihood $L(\pi, \beta)$ of less than $10^{-9}$ after one EM iteration. The log-likelihood for a given set of parameters is given by $L(\pi, \beta) = \Sigma_t log\, p(r_t)$. The EM iterative procedure was repeated with 10 different initializations of the parameters to avoid local minima.

Note that for monkey and rat data, since we observed large variations in performance across sessions, the model based its choices on session-gain-modulated evidence $\bar{S}_{ti} = \gamma_t S_{ti}$ instead of raw evidence $S_{ti}$ (this had no impact for the deterministic variant since $\bar{S}_{ti}$ and $S_{ti}$ always have the same sign). We fitted the model from individual subject responses either with lapses $\pi_L$ and $\pi_R$ as free parameters, or fixed to $\pi_L = \pi_R = 0.01$. Figures in the main manuscript correspond to the deterministic snapshot model with fixed lapses. We also studied variants of the snapshot model where decisions

in each trial are based on $K$ attended samples, that is depends on $(S_{ti}, ..S_{t,i+K-1})$ with $1 \leq K \leq n-1$ and $1 \leq i \leq n - K + 1$ is the first attended sample. In the deterministic case, the choice is directly determined by the sign of the sum of the signed evidence for the attended samples. In the non-deterministic case, the evidence for the attended samples are weighted and passed through a sigmoid: $H_i(S_t) = \sigma\left(\Sigma_{k \in [1...K]} \beta_{ki} S_{t,i+k-1}\right)$. The model with a single attended sample presented above is equivalent to this extended model when using $K = 1$. At the other end, using $K = n$ corresponds to the temporal integration model (without the lateral bias).

## Extrema-detection model

In the extrema-detection model, a choice is selected according to the first sample in the sequence whose absolute evidence value reaches a certain threshold $\theta$, that is $p(r_t|\theta) = H(m_{ti})$, $|m_{ti}| \geq \theta$, $|m_{tj}| < \theta$ for all $j < i$. Here, $m_{ti}$ is the sample evidence corrupted by sensory noise $\varepsilon_{ti}$ which is distributed normally with variance $\sigma^2$ : $m_{ti} = S_{ti} + \varepsilon_{ti}$ with $\varepsilon_{ti} \sim N(0, \sigma^2)$. $H$ is the step function. If the stimulus sequence ends and no sample has reached the threshold, then the decision is taken at chance. As described in *Waskom and Kiani, 2018*, the probability for a rightward choice at trial $t$ can be expressed as:

$$p(r_t) = \Sigma_{i \leq n} \Phi\left(\frac{S_{ti} - \theta}{\sigma}\right) \Pi_{j<i}\left(1 - \Phi\left(\frac{S_{tj} - \theta}{\sigma}\right) - \Phi\left(\frac{-S_{tj} - \theta}{\sigma}\right)\right) +$$
$$\frac{1}{2}\Pi_{j \leq n}\left(1 - \Phi\left(\frac{S_{tj} - \theta}{\sigma}\right) - \Phi\left(\frac{-S_{tj} - \theta}{\sigma}\right)\right)$$

where $\Phi$ is the cumulative normal distribution. We also included the possibility for left and right lapses with probability $\pi_L$ and $\pi_R$. Following *Stine et al., 2020*, we explored an alternative default rule called 'last sample' rule: if the stimulus extinguishes and the threshold has not been reached, then the decision is based on the (noisy) last sample rather than simply by chance. This changes the equation describing the probability for rightward choices to:

$$p(r_t) = \Sigma_{i<n} \Phi\left(\frac{S_{ti} - \theta}{\sigma}\right) \Pi_{j<i}\left(1 - \Phi\left(\frac{S_{tj} - \theta}{\sigma}\right) - \Phi\left(\frac{-S_{tj} - \theta}{\sigma}\right)\right) +$$
$$\Phi\left(\frac{S_{tn}}{\sigma}\right) \Pi_{j<n}\left(1 - \Phi\left(\frac{S_{tj} - \theta}{\sigma}\right) - \Phi\left(\frac{-S_{tj} - \theta}{\sigma}\right)\right)$$

We also explored a variant of the model where the threshold changes on every sample, that is the equations above are changed by substituting $\theta$ with $\theta_i$. As for the snapshot model, we used the session-gain-modulated evidence $\bar{S}_{ti}$ instead of raw evidence $S_{ti}$ for fitting the model to monkey and rat data. The four parameters of the model $(\theta, \sigma, \pi_L, \pi_R)$ were estimated from each subject data by maximizing the log-likelihood with interior-point algorithm (function *fmincon* in Matlab) and 10 different initializations of the parameters. (In the varying-threshold variant, there are $n + 3$ parameters which are estimated similarly.)

## Model validation and model comparison

Psychophysical kernels were obtained from subject and simulated data by running a logistic regression model: $p(r_t) = \sigma(\beta_0 + \Sigma_i \beta_i S_{ti})$. Standard errors of the weights $\beta_i$ were obtained from the Laplace approximation. For psychometric curves, we first defined the weighted stimulus evidence $T_t$ at trial $t$ as the session-modulated weighted sum of signed sample evidence; with the weights obtained from the logistic regression model above $T_t = \gamma_t \Sigma_i \beta_i S_{ti}$. We then divided the total stimulus evidence into 50 quantiles (10 for human subjects) and computed the psychometric curve as the proportion of rightward choices for each quantile.

The boundary performance for the snapshot and extrema-detection models corresponds to the best choice accuracy out of all the parameterizations for each model. In the snapshot model, the boundary performance corresponds to the deterministic version with no-lapse, where the attended sample is always the sample $i^*$ whose sign better predicts the stimulus category over all animal trials, that is $\pi_{i^*} = 1$ and $\pi_i = 0$ if $i \neq i^*$. For the extrema-detection model, the boundary performance corresponds to the lapse-free model with no sensory noise ($\sigma = 0$) and a certain value for threshold $\theta$ that is identified for each subject by simple parameter search.

Finally, model selection was performed using the AIC $AIC = 2p - 2L_{ML}$ , where $p$ is the number of model parameters and $L_{ML}$ is the likelihood evaluated at maximum likelihood parameters.

## Analysis of majority-driven choices

We selected for each animal the subset of trials corresponding to when the largest evidence sample was at odds with the total stimulus evidence (disagree trials), that is where $sign\left(S_{tj}, |S_{tj}| \geq |S_{ti}| \lor i\right) \neq sign\left(\Sigma_i S_{ti}\right)$. For this subset of trials, we computed the proportion of animal choices that were aligned with the overall stimulus evidence. We repeated the analysis for simulated data from the integration and extrema-detection models. We computed error bars following a parametric bootstrap procedure: for each bootstrap, we simulated the model (integration/extrema-detection) using parameters sampled from their posterior distribution (based on the Laplace approximation). We then applied the analysis on disagree trials for simulated data, and used these bootstrap values to define confidence intervals.

## Subjective weighting analysis

In order to estimate the impact of each sample on the animal choice as a function of sample evidence, we built and estimated the following statistical model:

$$p\left(r_t = A\right) \;=\; \sigma\left(\beta_0 + \gamma_t \Sigma_{i \in [1...n]} \beta_i f\left(S_{ti}\right)\right)$$

As can be seen, this model is equivalent to the temporal integration model under the assumption that $f$ is a linear function. Rather, here we wanted to estimate the function $f$ (as well as the session gain $\gamma_t$ , lateral bias $\beta_0$, and sensory weight $\beta_i$). Including the session gain was necessary for estimating $f$ accurately from the monkey and rat behavioral data, since the distribution of pulse strength $S_{ti}$ was varied across sessions and could otherwise induce a confound. We assumed that $f$ is an odd function, that is $f\left(-S_{ti}\right) = -f\left(S_{ti}\right)$. This equation takes the form of a GUM and was fitted using the Laplace approximation method as described in *Adam and Hyafil, 2020*. In the monkey experiment, sample evidence could take only a finite number of values, so $f$ was simply estimated over these values. In the human experiment, sample evidence could take continuous values. In this case, we defined a Gaussian Process prior over $f$ with squared exponential kernel with length scale 0.1 and variance 1.

## Integration of early and late evidence

We designed a new analysis tool to characterize the statistical mapping from the multidimensional stimulus space $S_t = (S_{t1}, ...S_{tn}) \in \mathfrak{R}^n$ onto binary choices $r_t \in [0, 1]$. We first collapsed the stimulus sequence $S_t$ onto the two-dimensional space defined by early evidence $E_t$ and late evidence $L_t$ defined by $E_t = \gamma_t \Sigma_{1 \leq i \leq [n/2]} \beta_i S_{ti}$ and $L_t = \gamma_t \Sigma_{[n/2]+1 \leq i \leq n} \beta_i S_{ti}$ , where the weights $\beta_i$ and session gains $\gamma_t$ correspond to parameters estimated from the temporal integration model (session gains were omitted for human participants). Next we plotted the integration map which represents the probability for rightward choices as a function of $(E_t, L_t)$. The map was obtained by smoothing data points with a two-dimensional Gaussian kernel. More specifically, for each pair value $(E, L)$, we selected the trials whose early and late evidence values $E_t$ and $L_t$ fell within a certain distance to $(E, L)$, that is $d_t = dist\left((E, L)(E_t, L_t)\right) < 2$. We then computed the proportion of rightward choices for the selected trials, with a weight for each trial depending on the distance to the pair value $w_t = N\left((E_t, L_t); (E, L), 0.1^2 I\right)$. Because the space $(E, L)$ was not sampled uniformly during the experiment, we represent the density of trials by brightness. For each subject, we obtained integration maps both from subject data and from model simulations. For each model, we computed the Pearson correlation between the maps obtained from the corresponding simulation and from the subject data. We tested the significance of correlation measures between models by using a bootstrapping procedure: we calculated the correlation measure $r$ from 100 bootstraps for each model and participant, and then performed an unpaired $t$-test between bootstrapped $r$.

Next, we analyzed the conditional psychometric curves, that is the psychometric curves for the early evidence conditioned on the value of late evidence, which correspond to vertical cuts in the integration map. To do so, we first binned late evidence $L_t$ by bins of width 0.5. Conditional psychometric curve represents the probability of rightward choices as a function of early evidence $E_t$, separately for each late evidence bin. For each late evidence bin, we also estimated the corresponding bias $\beta$, left lapse $\pi_L$ and right lapse $\pi_R$ by fitting the following function on the corresponding subset of trials:

$$p\left(r_t\right) = \pi_R + \left(1 - \pi_R - \pi_L\right) \sigma\left(\beta E_t\right)$$

## Analysis of LIP neuron activity

We analyzed the activity of 82 LIP neurons recorded over 43 sessions of the motion discrimination tasks (**Yates et al., 2017**). We applied the following procedure to extract the integration map for LIP neurons. For each neuron $n$, we computed the spike count $s_t^{(n)}$ in a window of 500 ms width following each stimulus offset, which is where LIP neurons were found to have maximal selectivity to motion evidence from the entire pulse sequence (**Yates et al., 2017**). We then applied a Poisson GLM $E\left(s_t^{(n)}\right) = exp\left(w_0^{(n)} + \Sigma_i w_i^{(n)} S_{ti}\right)$ for each neuron $n$ to extract the impact of each sample $i$ on the individual neural spike count $w_i^{(n)}$. For each trial $t$, we used these weights to compute the neuron-weighted early and late evidence defined by and $E_t^{(n)} = \Sigma_{1 \leq i \leq 3} w_i^{(n)} S_{ti}$ and $L_t^{(n)} = \Sigma_{4 \leq i \leq 7} w_i^{(n)} S_{ti}$. Note that this weighting converts the evidence onto the space defined by the preferred direction of the neuron, such that positive evidence signals evidence toward the preferred direction and negative evidence signals evidence toward the anti-preferred direction. We then merged the vectors for normalized spike counts $\bar{s}_t^{(n)} = s_t^{(n)}/exp\left(w_0^{(n)}\right)$, early evidence $E_t^{(n)}$ and late evidence $L_t^{(n)}$ across all neurons. The normalized spike counts were binned by values of early and late evidence (bin width: 0.02), and the average over each bin was computed after convolving with a two-dimensional Gaussian kernel of width 0.1. The neural integration map represents the average normalized activity per bin.

Simulations of spiking data for the integration and non-integration models were proceeded as follows. First, the neural integration model corresponds to linear summing with neuron-specific weights which are then passed through an exponential nonlinearity; the spike counts for each trial are generated using a Poisson distribution whose rate is equal to the nonlinear output (**Figure 4—figure supplement 3a**, top). This corresponds exactly to the generative process of the Poisson GLM described above. For the extrema-detection model (**Figure 4—figure supplement 3a**, middle), we hypothesized that LIP activity would only be driven by the sample that reaches the threshold (and dictates the animal response). To this end, we first simulated the behavioral extrema-detection model for all trials, using parameters $\left(\theta, \sigma, \pi_L, \pi_R\right)$ fitted from the corresponding animal, to identify which sample $i$ reaches the subject-specific threshold. We then assumed that the spiking activity of the neuron would follow the stimulus value at sample $i$ $S_{ti}$ (signed by the preferred direction of the neuron $p^{(n)}$ through):

$$E_{ED}\left(s_t^{(n)}\right) = \exp\left(w_0^{(n)} + p^{(n)} S_{ti} \Sigma_j w_j^{(n)}/2\right)$$

Again the spike count were generated from a Poisson distribution with rate $E_{ED}\left(s_t^{(n)}\right)$.

Finally, for the snapshot model (**Figure 4—figure supplement 3a**, bottom), we assumed that the neuron activity would merely reflect the sensory value of the only sample it would attend. We assumed that the probability mass function to attend each of the seven samples would be neuron specific, so we used the normalized weights of the Poisson GLM for that specific neuron as defining such probability (weights were signed by the neuron preferred direction so that the vast majority of weights were positive; negative weights were ignored). For each trial, we thus randomly sampled the attended sample $i$ using this probability mass function and then simulated the spike count $s_t^{(n)}$ from a Poisson distribution with rate

$$E_{Snapshot}\left(s_t^{(n)}\right) = \exp\left(w_0^{(n)} + p^{(n)} S_{ti} \Sigma_j w_j^{(n)}\right).$$

We simulated spiking activity for each neuron and for each integration and non-integration model, and then used simulated data to compute neural integration maps exactly as described above for the actual LIP neuron activity.

## Acknowledgements

The authors thank Jake Yates for sharing information regarding the monkey experimental data. The authors are supported by the Spanish State Research Agency (RYC-2017-23231 to AH), Spanish Ministry of Economy and Competitiveness together with the European Regional Development Fund grant SAF2015-70324-R (to JR), European Research Council grant ERC-2015-CoG-683209 (to JR), NIH grant R01EY017366 (to ACH and JWP), NIH BRAIN Initiative grant NS104899 (to JWP), and the Simons Collaboration on the Global Brain (SCGB AWD543027, JWP).

## Additional information

### Funding

| Funder | Grant reference number | Author |
|---|---|---|
| Agencia Estatal de Investigación | RYC-2017-23231 | Alexandre Hyafil |
| Ministerio de Economía y Competitividad | SAF2015-70324-R | Jaime de la Rocha |
| European Research Council | ERC-2015-CoG-683209 | Jaime de la Rocha |
| National Institutes of Health | R01EY017366 | Alexander C Huk Jonathan W Pillow |
| National Institutes of Health | NS104899 | Jonathan W Pillow |
| Simons Collaboration for the Global Brain | SCGB AWD543027 | Jonathan W Pillow |

The funders had no role in study design, data collection, and interpretation, or the decision to submit the work for publication.

### Author contributions

Alexandre Hyafil, Conceptualization, Data curation, Software, Formal analysis, Funding acquisition, Investigation, Visualization, Methodology, Writing - original draft, Writing - review and editing; Jaime de la Rocha, Investigation, Writing - review and editing; Cristina Pericas, Data curation, Investigation; Leor N Katz, Conceptualization, Data curation, Writing - review and editing; Alexander C Huk, Conceptualization, Data curation, Supervision, Funding acquisition, Investigation, Methodology, Writing - review and editing; Jonathan W Pillow, Conceptualization, Supervision, Funding acquisition, Investigation, Visualization, Methodology, Writing - review and editing

### Author ORCIDs

Alexandre Hyafil http://orcid.org/0000-0002-0566-651X
Jaime de la Rocha http://orcid.org/0000-0002-3314-9384
Leor N Katz http://orcid.org/0000-0002-2742-6533
Alexander C Huk http://orcid.org/0000-0003-1430-6935
Jonathan W Pillow http://orcid.org/0000-0002-3638-8831

### Ethics

Informed consent was obtained from all participants. The experiment with human participants was approved by the UPF ethics committee (approval 654 2013/5435/I from CEIm-Parc de Salut MAR). Rat experiments were approved by the local ethics committee of the University of Barcelona 658 (Comité d'Experimentació Animal, Barcelona, Spain, protocol number Ref 390/14). Monkey experiment: All experimental protocols were approved by The University of Texas Institutional Animal Care

and Use Committee (AUP-2012-00085, AUP-2015-00068) and in accordance with National Institutes of Health standards for care and use of laboratory animals.

### Decision letter and Author response
Decision letter https://doi.org/10.7554/eLife.84045.sa1
Author response https://doi.org/10.7554/eLife.84045.sa2

---

## Additional files

### Supplementary files
• MDAR checklist

### Data availability
All experimental data (behavioral and neural data in monkeys, behavioral data in rats and humans) and code to run the analysis are publicly available at https://github.com/ahyafil/TemporalIntegration (copy archieved at *Hyafil, 2023*).

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
