## [Editor Report]

This manuscript tests an important assumption about how sensory information is processed and used to guide motor choices. The widely held assumption is that sensory-motor circuits are capable of integrating evidence, but the validity and generality of this 'principle' have been recently questioned by studies suggesting that other computational operations may lead to similar psychophysical results, mimicking integration without actually performing it. This study makes a compelling case that the integration assumption was likely correct all along and that the model mimicry can be easily disambiguated by using appropriate sensory stimuli and task designs that permit rigorous analyses.

---

## [Decision Letter]

**Decision letter after peer review:**

Thank you for submitting your article "Temporal integration is a robust feature of perceptual decisions" for consideration by *eLife*. Your article has been reviewed by 2 peer reviewers, including Emilio Salinas as the Reviewing Editor and Reviewer #1, and the evaluation has been overseen by Michael Frank as the Senior Editor.

Essential revisions:

1) Reviewer 2 pointed out that the model used by the authors is one of a broader family of possible models. Consideration of this broader family, at least in the form of some discussion, would strengthen the results, or would at least situate them more clearly in the theoretical landscape. See the full recommendation of Reviewer 2 below.

*Reviewer #2 (Recommendations for the authors):*

I have one main suggestion. Otherwise, I think this is a great study.

A very general family of models of integration of a variable "x" in response to a noisy stimulus "s" would be:

tau. dx/dt = s(t). F(x,t) + G(x,t) where G(x,t) would represent any dynamics in the absence of stimulus and F(x,t) would weight evidence according to time and current level of integrated evidence. It is unclear to this reader if the results have placed any restrictions on the potential forms of G(x,t) – the suggestion is that it should be zero, but this has not been tested I think – nor on whether the function "F()" which they show depends on time, "t" could also depend on the level of integrated evidence, "x". Since the model only includes an F(t) but then contrasts with very different models, there is an implicit suggestion or assumption that F(x,t) = F(t) and G(x,t) = 0, but this assumption is not tested. It would be nice to see an explicit test or a clear statement on what limits or lack thereof must be present in this integration-like process. For example, a comparison with the urgency-gating model of Cisek (J Neurosci 2009) would be valuable, along with a discussion of whether the authors expect different results in free response paradigms.

Supp Figure 2: This needs a bit more explaining. Legend in A is different from the caption. In B, with Δ-AIC, shouldn't the AIC be based on the likelihood of the data-given model? So why is there a black curve for "data"? In fact, how are their 3 curves at all as a Δ-AIC is a difference between two values? Confusing. Perhaps this is just "AIC" and the black curve is the integration model, not the experimental data?

---

## [Author Response]

Essential revisions:Reviewer #2 (Recommendations for the authors):I have one main suggestion. Otherwise, I think this is a great study.A very general family of models of integration of a variable "x" in response to a noisy stimulus "s" would be:tau. dx/dt = s(t). F(x,t) + G(x,t) where G(x,t) would represent any dynamics in the absence of stimulus and F(x,t) would weight evidence according to time and current level of integrated evidence. It is unclear to this reader if the results have placed any restrictions on the potential forms of G(x,t) – the suggestion is that it should be zero, but this has not been tested I think – nor on whether the function "F()" which they show depends on time, "t" could also depend on the level of integrated evidence, "x". Since the model only includes an F(t) but then contrasts with very different models, there is an implicit suggestion or assumption that F(x,t) = F(t) and G(x,t) = 0, but this assumption is not tested. It would be nice to see an explicit test or a clear statement on what limits or lack thereof must be present in this integration-like process. For example, a comparison with the urgency-gating model of Cisek (J Neurosci 2009) would be valuable, along with a discussion of whether the authors expect different results in free response paradigms.

We thank the reviewer for pointing to this very important point. While the two non-integration models were relatively easy to formalize into generic generative models that can later be fitted to experimental data, the same does not hold for the integration model. As rightfully pointed out, the most general generative model of integration should include dependence of the drift term on both time and x, both for the stimulus-sensitive part F(x,t) and G(x,t). This wider class of models could thus accommodate time-varying sticky bounds theta(t) (i.e. F(theta(t),t)=G(theta(t),t)=0), leak or attractive dynamics (G(x,t) = λ*x) and more. Now, fitting such an immense class of models to experimental data becomes very challenging computationally and prone to overfitting (observed values consist of 500-50 000 binary observations). For this reason we decided to restrict to a much high-level description of the computation that does not characterize the functions G() or F(), and that should not be interpreted literally as a generative model of behavior but rather as a (first-order) statistical description for any of the generative integration model corresponding to the larger integration family. Thus, to connect with the reviewer’s public review, we do not claim that the non-monotonic weights of the logistic model correspond genuinely to a non-monotonic sensitivity F(t), but are rather the statistical linear description of more complex nonlinear phenomena. For example, non-monotonic kernels (both bell-shaped and U-shaped) have been described in previous studies and linked theoretically to either attractor dynamics (i.e. double well potential, G(x,t) = a.x^3 + b.x in Prat-Ortega et al., 2021; or time-dependent dynamics in Bronfman, Brezis, and Usher 2016) or divisive normalization (Keung et al. 2019 and 2020). Technically, our statistical approach corresponds to the first-order expansion of the Volterra series used to approximate any integration model (Neri, 2004) by a simple and fittable logistic model.

Importantly our objective in this paper was not to compare and identify these complex mechanisms at play during evidence integration, but rather to compare non-integration models against *any* integration strategy. We believe that our first-order statistical modelling of integration strategies is suited for this purpose.

References

Keung, W., Hagen, T.A. & Wilson, R.C. A divisive model of evidence accumulation explains uneven weighting of evidence over time. *Nat Commun* 11, 2160 (2020). https://doi.org/10.1038/s41467-020-15630-0

Keung, Waitsang, Todd A. Hagen, and Robert C. Wilson. 2019. “Regulation of Evidence Accumulation by Pupil-Linked Arousal Processes.” *Nature Human Behaviour* 3 (6): 636–45.

Neri, P. (2004). Estimation of nonlinear psychophysical kernels. *Journal of Vision*, *4*(2), 2. https://doi.org/10.1167/4.2.2

Prat-Ortega, G., Wimmer, K., Roxin, A. *et al.* Flexible categorization in perceptual decision making. *Nat Commun* 12, 1283 (2021). https://doi.org/10.1038/s41467-021-21501-z

We have added the following clarifications in the manuscript: “Departures from optimality in the accumulation process such as accumulation leak, categorization dynamics, sensory adaptation or sticky boundaries may however yield unequal weighting of the different samples (Yates et al. 2017; Brunton, Botvinick, and Brody 2013; Prat-Ortega et al. 2021; Bronfman, Brezis, and Usher 2016; Keung, Hagen, and Wilson 2020, 2019). To accommodate for these, we allowed the model to take any arbitrary weighting of the samples: *p*(*r* = *A*) = Φ(β0 + Σ*i*β*i*S*i*) (see Methods for details). The mapping from final accumulated evidence to choice was probabilistic, to account for the effects of noise from different sources in the decision-making process (Drugowitsch et al. 2016). Thus the model represented an approximate statistical description for any generative model relying on temporal integration of the stimulus evidence.” (Results, p7)

“The vector β*i*’s allowed to capture different shapes of psychophysical kernels (e.g. primacy effects, recency effects) which can emerge due to a variety of suboptimalities in the integration process (leak, attractor dynamics, sticky bounds, sensory after-effects, etc.) (Brunton, Botvinick, and Brody 2013; Yates et al. 2017; Prat-Ortega et al. 2021; Bronfman, Brezis, and Usher 2016; Keung, Hagen, and Wilson 2019, 2020). Technically, our statistical approach corresponds to the first-order expansion of the Volterra series used to approximate any integration model (Neri 2004) by a simple and fittable logistic model. In other words, because a more complete description of integration models would be computationally challenging to fit and prone to overfitting, we chose a statistical approximation in the form of the logistic model that captures the essence of any generative model that include temporal integration, i.e. weighted summation of stimulus evidence.” (Methods, p22).

Finally, we have expanded the discussion about free-response paradigms in the Discussion to include an explicit comparison with the urgency-gating model of Cisek et al. 2009 (in the previous version, we referred to Thura et al. 2012 which is a more thorough theoretical development of the urgency-gating model):

“Other studies have shown how integration and non-integration strategies can be disentangled in free reaction-time task paradigms. Specifically, different models make different predictions regarding how the total sample evidence presented before response time should vary with response time (Glickman and Usher 2019; Zuo and Diamond 2019). Glickman and Usher used these predictions to rule out non-integration strategies in a counting task in humans, and Zuo and Diamond found evidence for evidence integration to bound when rats discriminate textures using whisker touches (Zuo and Diamond 2019). Under strong urgency constraints, it has been proposed that decisions depend on very limited temporal integration of the stimulus by low-pass filtering of stimulus evidence (Cisek et al. 2009; Thura et al. 2012). However this suggestion cannot explain the fact that evidence presented early in the trial influences decisions taken later on (Winkel et al. 2014). Of note, the absence of integration seems a more viable strategy when the duration of the stimulus is controlled externally and the benefits of integrating in terms of accuracy might not compensate for its cognitive cost. In free-reaction time paradigms, waiting for a long sequence of samples and selecting its response based on a single sample does not seem a particularly efficient strategy. If the cognitive cost of integration is high, it is more beneficial to interrupt the stimulus sequence early with a rapid response. Indeed, fast or very fast (under 250 ms) perceptual decisions are very common (Uchida, Kepecs, and Mainen 2006; Zariwala et al. 2013; Stanford and Salinas 2021). Such rapid termination of the decision process have can been attributed either to urgency signals modulating the integration of stimulus evidence (Drugowitsch et al. 2012; Stanford and Salinas 2021) or to action initiation mechanisms that time the response after a specific time (e.g. one or two samples) following stimulus onset (Hernández-Navarro et al. 2021). Here, we have shown that even in paradigms where the stimulus duration is controlled by the experimenter, mammals often integrate sensory evidence over the entire stimulus.”

Supp Figure 2: This needs a bit more explaining. Legend in A is different from the caption.

Thank you for spotting multiple problems with this figure caption. Caption for panel A has been corrected and is now in agreement with the legend. (This figure is now labeled ‘Figure 2 figure supplement 2).

In B, with Δ-AIC, shouldn't the AIC be based on the likelihood of the data-given model? So why is there a black curve for "data"? In fact, how are their 3 curves at all as a Δ-AIC is a difference between two values? Confusing. Perhaps this is just "AIC" and the black curve is the integration model, not the experimental data?

There are actually 2x2 = 4 different curves for each of the different variants of the snapshot model (deterministic vs probabilistic, and free vs fixed lapse parameters), but two curves (probabilistic-free and probabilistic free) are virtually identical. Each curve corresponds to Δ-AIC that measures the difference in AIC for this specific variant of the snapshot model w.r.t the integration model (i.e. the integration model always corresponds to Δ-AIC = 0). So in essence the panel compares 4 x 6 = 24 variants of the snapshot model with the integration model, and the integration model always wins (for both animals). We have clarified this point in the caption:

“AIC difference between each of the four variants of the snapshot model and the integration model. Legend as in A (full/dashed lines for fixed/free lapse parameters; black/blue curves for probabilistic/deterministic variants). Note that the probabilistic variant with either fixed or free lapses provide virtually indistinguishable values. Positive values indicate that the snapshot model provides a worse fit than the integration model.”